# Global Convergence of Direct Policy Search for State-Feedback $\mathcal{H}_\infty$ Robust Control: A Revisit of Nonsmooth Synthesis with Goldstein Subdifferential

**Xingang Guo,    Bin Hu**
Department of Electrical and Computer Engineering
Coordinated Science Laboratory
University of Illinois at Urbana-Champaign
{xingang2,binhu7}@illinois.edu

## Abstract

Direct policy search has been widely applied in modern reinforcement learning and continuous control. However, the theoretical properties of direct policy search on nonsmooth robust control synthesis have not been fully understood. The optimal $\mathcal{H}_\infty$ control framework aims at designing a policy to minimize the closed-loop $\mathcal{H}_\infty$ norm, and is arguably the most fundamental robust control paradigm. In this work, we show that direct policy search is guaranteed to find the global solution of the robust $\mathcal{H}_\infty$ state-feedback control design problem. Notice that policy search for optimal $\mathcal{H}_\infty$ control leads to a constrained nonconvex nonsmooth optimization problem, where the nonconvex feasible set consists of all the policies stabilizing the closed-loop dynamics. We show that for this nonsmooth optimization problem, all Clarke stationary points are global minimum. Next, we identify the coerciveness of the closed-loop $\mathcal{H}_\infty$ objective function, and prove that all the sublevel sets of the resultant policy search problem are compact. Based on these properties, we show that Goldstein's subgradient method and its implementable variants can be guaranteed to stay in the nonconvex feasible set and eventually find the global optimal solution of the $\mathcal{H}_\infty$ state-feedback synthesis problem. Our work builds a new connection between nonconvex nonsmooth optimization theory and robust control, leading to an interesting global convergence result for direct policy search on optimal $\mathcal{H}_\infty$ synthesis.

## 1   Introduction

Reinforcement learning (RL) has achieved impressive performance on many continuous control tasks [59, 40], and policy optimization is one of the main workhorses for such applications [18, 65, 58, 60]. Recently, there have been extensive research efforts studying the global convergence properties of policy optimization methods on benchmark control problems including linear quadratic regulator (LQR) [21, 7, 41, 70, 44, 22, 29], stabilization [52, 51], linear robust/risk-sensitive control [73, 72, 26, 74, 75, 12], Markov jump linear quadratic control [32, 31, 33, 55], Lur'e system control [53], output feedback control [20, 77, 39, 17, 16, 43, 76], and dynamic filtering [68]. For all these benchmark problems, the objective function in the policy optimization formulation is always differentiable over the entire feasible set, and the existing convergence theory heavily relies on this fact. Consequently, an important open question remains whether direct policy search can enjoy similar global convergence properties when applied to the famous $\mathcal{H}_\infty$ control problem whose objective function can be non-differentiable over certain points in the policy space [1–3, 28, 9, 13, 48]. Different from LQR which considers stochastic disturbance sequences, $\mathcal{H}_\infty$ control directly addresses the worst-case disturbance, and provides arguably the most fundamental robust control paradigm [78, 19, 62, 4, 15, 23]. Regarding

36th Conference on Neural Information Processing Systems (NeurIPS 2022).

the connection with RL, it has also been shown that $\mathcal{H}_\infty$ control can be applied to stabilize the training of adversarial RL schemes in the linear quadratic setup [72, Section 5]. Given the fundamental importance of $\mathcal{H}_\infty$ control, we view it as an important benchmark for understanding the theoretical properties of direct policy search in the context of robust control and adversarial RL. In this work, we study and prove the global convergence properties of direct policy search on the $\mathcal{H}_\infty$ state-feedback synthesis problem.

The objective of the $\mathcal{H}_\infty$ state-feedback synthesis is to design a linear state-feedback policy that stabilizes the closed-loop system and minimizes the $\mathcal{H}_\infty$ norm from the disturbance to a performance signal at the same time. The design goal is also equivalent to synthesizing a state-feedback policy that minimizes a quadratic cost subject to the worst-case disturbance. We will present the problem formulation for the $\mathcal{H}_\infty$ state-feedback synthesis and discuss such connections in Section 2. Essentially, $\mathcal{H}_\infty$ state-feedback synthesis can be formulated as a constrained policy optimization problem $\min_{K \in \mathcal{K}} J(K)$, where the decision variable $K$ is a matrix parameterizing the linear state-feedback policy, the objective function $J(K)$ is the closed-loop $\mathcal{H}_\infty$-norm for given $K$, and the feasible set $\mathcal{K}$ consists of all the linear state-feedback policies stabilizing the closed-loop dynamics. Notice that the feasible set for the $\mathcal{H}_\infty$ state-feedback control problem is the same as the nonconvex feasible set for the LQR policy search problem [21, 7]. However, the objective function $J(K)$ for the $\mathcal{H}_\infty$ control problem can be non-differential over certain feasible points, introducing new difficulty to direct policy search. There has been a large family of nonsmooth $\mathcal{H}_\infty$ policy search algorithms developed based on the concept of Clarke subdifferential [1–3, 28, 9, 13]. However, a satisfying global convergence theory is still missing from the literature. Our paper bridges this gap by making the following two contributions.

1. We show that all Clarke stationary points for the $\mathcal{H}_\infty$ state-feedback policy search problem are also global minimum.
2. We identify the coerciveness of the $\mathcal{H}_\infty$ cost function and use this property to show that Goldstein's subgradient method [25] and its implementable variants [71, 14, 9, 10, 37, 38] can be guaranteed to stay in the nonconvex feasible set of stabilizing policies during the optimization process and eventually find the global optimal solution of the $\mathcal{H}_\infty$ state-feedback control problem. Finite-time complexity bounds for finding $(\delta, \epsilon)$-stationary points are also provided.

Our work sheds new light on the theoretical properties of policy optimization methods on $\mathcal{H}_\infty$ control problems, and serves as a meaningful initial step towards a general global convergence theory of direct policy search on nonsmooth robust control synthesis.

Finally, it is worth clarifying the differences between $\mathcal{H}_\infty$ control and mixed $\mathcal{H}_2/\mathcal{H}_\infty$ design. For mixed $\mathcal{H}_2/\mathcal{H}_\infty$ control, the objective is to design a stabilizing policy that minimizes an $\mathcal{H}_2$ performance bound and satisfies an $\mathcal{H}_\infty$ constraint at the same time [24, 36, 34, 47]. In other words, mixed $\mathcal{H}_2/\mathcal{H}_\infty$ control aims at improving the average $\mathcal{H}_2$ performance while "maintaining" a certain level of robustness by keeping the closed-loop $\mathcal{H}_\infty$ norm to be smaller than a pre-specified number. In contrast, $\mathcal{H}_\infty$ control aims at "improving" the system robustness and the worst-case performance via achieving the smallest closed-loop $\mathcal{H}_\infty$ norm. In [73], it has been shown that the natural policy gradient method initialized from a policy satisfying the $\mathcal{H}_\infty$ constraint can be guaranteed to maintain the $\mathcal{H}_\infty$ requirement during the optimization process and eventually converge to the optimal solution of the mixed design problem. However, notice that the objective function for the mixed $\mathcal{H}_2/\mathcal{H}_\infty$ control problem is still differentiable over all the feasible points, and hence the analysis technique in [73] cannot be applied to our $\mathcal{H}_\infty$ control setting. More discussions on the connections and differences between these two problems will be given in the supplementary material.

## 2 Problem Formulation and Preliminaries

### 2.1 Notation

The set of $p$-dimensional real vectors is denoted as $\mathbb{R}^p$. For a matrix $A$, we use the notation $A^\mathsf{T}$, $\|A\|$, $\operatorname{tr} A$, $\sigma_{\min}(A)$, $\|A\|_2$, and $\rho(A)$ to denote its transpose, largest singular value, trace, smallest singular value, Frobenius norm, and spectral radius, respectively. When a matrix $P$ is negative semidefinite (definite), we will use the notation $P \preceq (\prec)0$. When $P$ is positive semidefinite (definite), we use the notation $P \succeq (\succ)0$. Consider a (real) sequence $\mathbf{u} := \{u_0, u_1, \cdots\}$ where $u_t \in \mathbb{R}^{n_u}$ for all $t$. This

sequence is said to be in $\ell_2^{n_u}$ if $\sum_{t=0}^{\infty} \|u_t\|^2 < \infty$ where $\|u_t\|$ denotes the standard (vector) 2-norm of $u_t$. In addition, the 2-norm for $\mathbf{u} \in \ell_2^{n_u}$ is defined as $\|\mathbf{u}\|^2 := \sum_{t=0}^{\infty} \|u_t\|^2$.

## 2.2 Problem statement: $\mathcal{H}_\infty$ state-feedback synthesis and a policy optimization formulation

We consider the following linear time-invariant (LTI) system

$$x_{t+1} = Ax_t + Bu_t + w_t, \; x_0 = 0 \tag{1}$$

where $x_t \in \mathbb{R}^{n_x}$ is the state, $u_t \in \mathbb{R}^{n_u}$ is the control action, and $w_t \in \mathbb{R}^{n_w}$ is the disturbance. We have $A \in \mathbb{R}^{n_x \times n_x}$, $B \in \mathbb{R}^{n_x \times n_u}$, and $n_w = n_x$. We denote $\mathbf{x} := \{x_0, x_1, \cdots\}$, $\mathbf{u} := \{u_0, u_1, \cdots\}$, and $\mathbf{w} := \{w_0, w_1, \cdots\}$. The initial condition is fixed as $x_0 = 0$. The objective of $\mathcal{H}_\infty$ control is to choose $\{u_t\}$ to minimize the quadratic cost $\sum_{t=0}^{\infty}(x_t^\mathsf{T} Q x_t + u_t^\mathsf{T} R u_t)$ in the presence of the worst-case $\ell_2$ disturbance satisfying $\|\mathbf{w}\| \leq 1$. In this paper, the following assumption is adopted.

**Assumption 1.** *The matrices $Q$ and $R$ are positive definite. The matrix pair $(A, B)$ is stabilizable.*

In $\mathcal{H}_\infty$ control, $\{w_t\}$ is considered to be the worst-case disturbance satisfying the $\ell_2$ norm bound $\|\mathbf{w}\| \leq 1$, and can be chosen in an adversarial manner. This is different from LQR which makes stochastic assumptions on $\{w_t\}$. Without loss of generality, we have chosen the $\ell_2$ upper bound on $\mathbf{w}$ to be 1. In principle, we can formulate the $\mathcal{H}_\infty$ control problem with any arbitrary $\ell_2$ upper bound on $\mathbf{w}$, and there is no technical difference. We will provide more explanations on this fact in the supplementary material. Therefore, $\mathcal{H}_\infty$ control can be formulated as the following minimax problem

$$\min_{\mathbf{u}} \max_{\mathbf{w}:\|\mathbf{w}\| \leq 1} \sum_{t=0}^{\infty}(x_t^\mathsf{T} Q x_t + u_t^\mathsf{T} R u_t) \tag{2}$$

Under Assumption 1, it is well known that the optimal solution for (2) can be achieved using a linear state-feedback policy $u_t = -Kx_t$ (see [4]). Given any $K$, the LTI system (1) can be rewritten as

$$x_{t+1} = (A - BK)x_t + w_t, \; x_0 = 0. \tag{3}$$

Now we define $z_t = (Q + K^\mathsf{T} RK)^{\frac{1}{2}} x_t$. We have $\|z_t\|^2 = x_t^\mathsf{T}(Q + K^\mathsf{T} RK)x_t = x_t^\mathsf{T} Q x_t + u_t^\mathsf{T} R u_t$. We denote $\mathbf{z} := \{z_0, z_1, \cdots\}$. If $\mathbf{x} \in \ell_2^{n_x}$, then we have $\|\mathbf{z}\|^2 = \sum_{t=0}^{\infty}(x_t^\mathsf{T} Q x_t + u_t^\mathsf{T} R u_t) < +\infty$. Therefore, the closed-loop LTI system (3) can be viewed as a linear operator mapping any disturbance sequence $\{w_t\}$ to another sequence $\{z_t\}$. We denote this operator as $G_K$, where the subscript highlights the dependence of this operator on $K$. If $K$ is stabilizing, i.e. $\rho(A - BK) < 1$, then $G_K$ is bounded in the sense that it maps any $\ell_2$ sequence $\mathbf{w}$ to another sequence $\mathbf{z}$ in $\ell_2^{n_x}$. For any stabilizing $K$, the $\ell_2 \to \ell_2$ induced norm of $G_K$ can be defined as:

$$\|G_K\|_{2\to2} := \sup_{0 \neq \|\mathbf{w}\| \leq 1} \frac{\|\mathbf{z}\|}{\|\mathbf{w}\|} \tag{4}$$

Since $G_K$ is a linear operator, it is straightforward to show

$$\|G_K\|_{2\to2}^2 := \max_{\mathbf{w}:\|\mathbf{w}\| \leq 1} \sum_{t=0}^{\infty} x_t^\mathsf{T}(Q + K^\mathsf{T} RK)x_t = \max_{\mathbf{w}:\|\mathbf{w}\| \leq 1} \sum_{t=0}^{\infty}(x_t^\mathsf{T} Q x_t + u_t^\mathsf{T} R u_t).$$

Therefore, the minimax optimization problem (2) can be rewritten as the policy optimization problem: $\min_{K \in \mathcal{K}} \|G_K\|_{2\to2}^2$, where $\mathcal{K}$ is the set of all linear state-feedback stabilizing policies, i.e. $\mathcal{K} = \{K \in \mathbb{R}^{n_u \times n_x} : \rho(A - BK) < 1\}$. In the robust control literature [1–3, 28, 9, 13], it is standard to drop the square in the cost function and just reformulate (2) as $\min_{K \in \mathcal{K}} \|G_K\|_{2\to2}$. This is exactly the policy optimization formulation for $\mathcal{H}_\infty$ state-feedback control. The main reason why this problem is termed as $\mathcal{H}_\infty$ state-feedback control is that in the frequency domain, $G_K$ can be viewed as a transfer function which lives in the Hardy $\mathcal{H}_\infty$ space and has an $\mathcal{H}_\infty$ norm being exactly equal to $\|G_K\|_{2\to2}$. Applying the frequency-domain formula for the $\mathcal{H}_\infty$ norm, we can calculate $\|G_K\|_{2\to2}$ as

$$\|G_K\|_{2\to2} = \sup_{\omega \in [0,2\pi]} \lambda_{\max}^{1/2}\big((e^{-j\omega}I - A + BK)^{-\mathsf{T}}(Q + K^\mathsf{T} RK)(e^{j\omega}I - A + BK)^{-1}\big), \tag{5}$$

where $I$ is the identity matrix, and $\lambda_{\max}$ denotes the largest eigenvalue of a given symmetric matrix. Therefore, eventually the $\mathcal{H}_\infty$ state-feedback control problem can be formulated as

$$\min_{K \in \mathcal{K}} J(K), \tag{6}$$

where $J(K)$ is equal to the $\mathcal{H}_\infty$ norm specified by (5). Classical $\mathcal{H}_\infty$ control theory typically solves (6) via introducing extra Lyapunov variables and reparameterizing the problem into a higher-dimensional convex domain over which convex optimization algorithms can be applied [78, 19, 6]. In this paper, we revisit (6) as a benchmark for direct policy search, and discuss how to search the optimal solution of (6) in the policy space directly. Applying direct policy search to address (6) leads to a nonconvex nonsmooth optimization problem. A main technical challenge is that the objective function (5) can be non-differentiable over some important feasible points [1–3, 28, 9, 13].

### 2.3 Direct policy search: A nonsmooth optimization perspective

Now we briefly review several key facts known for the $\mathcal{H}_\infty$ policy optimization problem (6).

**Proposition 1.** *The set $\mathcal{K} = \{K : \rho(A - BK) < 1\}$ is open. In general, it can be unbounded and nonconvex. The cost function* (5) *is continuous and nonconvex in $K$.*

See [21, 8] for some related proofs. We have also included more explanations in the supplementary material. An immediate consequence is that (6) becomes a nonconvex optimization problem. Another important fact is that the objective function (5) is also nonsmooth. As a matter of fact, (5) is subject to two sources of nonsmoothness. Based on (5), we can see that the largest eigenvalue for a fixed frequency $\omega$ is nonsmooth, and the optimization step over $\omega \in [0, 2\pi]$ is also nonsmooth. As a matter of fact, the $\mathcal{H}_\infty$ objective function (5) can be non-differentiable over important feasible points, e.g. optimal points. Fortunately, it is well known[1] that the $\mathcal{H}_\infty$ objective function (5) has the following desired property so it is Clarke subdifferentiable.

**Proposition 2.** *The $\mathcal{H}_\infty$ objective function* (5) *is locally Lipschitz and subdifferentially regular over the stabilizing feasible set $\mathcal{K}$.*

Recall that $J : \mathcal{K} \to \mathbb{R}$ is locally Lipschitz if for any bounded $S \subset \mathcal{K}$, there exists a constant $L > 0$ such that $|J(K) - J(K')| \le L\|K - K'\|_2$ for all $K, K' \in S$. Based on Rademacher's theorem, a locally Lipschitz function is differentiable almost everywhere, and the Clarke subdifferential is well defined for all feasible points. Formally, the Clarke subdifferential is defined as

$$\partial_C J(K) := \text{conv}\{\lim_{i \to \infty} \nabla J(K_i) : K_i \to K, \ K_i \in \text{dom}(\nabla J) \subset \mathcal{K}\} \tag{7}$$

where conv denotes the convex hull. Then we know that the Clarke subdifferential for the $\mathcal{H}_\infty$ objective function (5) is well defined for all $K \in \mathcal{K}$. We say that $K$ is a Clarke stationary point if $0 \in \partial_C J(K)$. The following fact is also well known.

**Proposition 3.** *If $K$ is a local min of $J$, then $0 \in \partial_C J(K)$ and $K$ is a Clarke stationary point.*

Under Assumption 1, it is well known that there exists $K^* \in \mathcal{K}$ achieving the minimum of (6). Since $\mathcal{K}$ is an open set, $K^*$ has to be an interior point of $\mathcal{K}$ and hence $K^*$ has to be a Clarke stationary point. In Section 3, we will prove that any Clarke stationary points for (6) are actually global minimum.

Now we briefly elaborate on the subdifferentially regular property stated in Proposition 2. For any given direction $d$ (which has the same dimension as $K$), the generalized Clarke directional derivative of $J$ is defined as

$$J^\circ(K, d) := \lim_{K' \to K} \sup_{t \searrow 0} \frac{J(K' + td) - J(K')}{t}. \tag{8}$$

In contrast, the (ordinary) directional derivative is defined as follows (when existing)

$$J'(K, d) := \lim_{t \searrow 0} \frac{J(K + td) - J(K)}{t}. \tag{9}$$

---

[1]We cannot find a formal statement of Proposition 2 in the literature. However, based on our discussion with other researchers who have worked on nonsmooth $\mathcal{H}_\infty$ synthesis for long time, this fact is well known and hence we do not claim any credits in deriving this result. As a matter of fact, although not explicitly stated, the proof of Proposition 2 is hinted in the last paragraph of [2, Section III] given the facts that the $\mathcal{H}_\infty$ norm is a convex function over the Hardy $\mathcal{H}_\infty$ space (which is a Banach space) and the mapping from $K \in \mathcal{K}$ to the (infinite-dimensional) Hardy $\mathcal{H}_\infty$ space is strictly differentiable. For completeness, a simple proof of Proposition 2 based on Clarke's chain rule [11] is included in the supplementary material.

In general, the Clarke directional derivative can be different from the (ordinary) directional derivative. Sometimes the ordinary directional derivative may not even exist. The objective function $J(K)$ is subdifferentially regular if for every $K \in \mathcal{K}$, the ordinary directional derivative always exists and coincides with the generalized one for every direction, i.e. $J'(K, d) = J^\circ(K, d)$. The most important consequence of the subdifferentially regular property is given as follows.

**Corollary 1.** *Suppose $K^\dagger \in \mathcal{K}$ is a Clarke stationary point for $J$. If $J$ is subdifferentially regular, then the directional derivatives $J'(K^\dagger, d)$ are non-negative for all $d$.*

See [56, Theorem 10.1] for related proofs and more discussions. Notice that having non-negative directional derivatives does not mean that the point $K^\dagger$ is a local minimum. Nevertheless, the above fact will be used in our main theoretical developments. Now we briefly summarize two key difficulties in establishing a global convergence theory for direct policy search on the $\mathcal{H}_\infty$ state-feedback control problem (6). First, it is unclear whether the direct policy search method will get stuck at some local minimum. Second, it is challenging to guarantee the direct policy search method to stay in the nonconvex feasible set $\mathcal{K}$ during the optimization process. Since $\mathcal{K}$ is nonconvex, we cannot use a projection step to maintain feasibility. Our main results will address these two issues.

### 2.4 Goldstein subdifferential

Generating a good descent direction for nonsmooth optimization is not trivial. Many nonsmooth optimization algorithms are based on the concept of Goldstein subdifferential [25]. Before proceeding to our main result, we briefly review this concept here.

**Definition 1** (Goldstein subdifferential). *Suppose $J$ is locally Lipschitz. Given a point $K \in \mathcal{K}$ and a parameter $\delta > 0$, the Goldstein subdifferential of $J$ at $K$ is defined to be the following set*

$$\partial_\delta J(K) := \operatorname{conv} \left\{ \cup_{K' \in \mathbb{B}_\delta(K)} \partial_C J(K') \right\}, \tag{10}$$

*where $\mathbb{B}_\delta(K)$ denotes the $\delta$-ball around $K$. The above definition implicitly requires $\mathbb{B}_\delta(K) \subset \mathcal{K}$.*

Based on the above definition, one can further define the notion of $(\delta, \epsilon)$-stationarity. A point $K$ is said to be $(\delta, \epsilon)$-stationary if $\operatorname{dist}(0, \partial_\delta J(K)) \le \epsilon$. It is well-known that the minimal norm element of the Goldstein subdifferential generates a good descent direction. This fact is stated as follows.

**Proposition 4** ([25]). *Let $F$ be the minimal norm element in $\partial_\delta J(K)$. Suppose $K - \alpha F/\|F\|_2 \in \mathcal{K}$ for any $0 \le \alpha \le \delta$. Then we have*

$$J(K - \delta F/\|F\|_2) \le J(K) - \delta \|F\|_2. \tag{11}$$

The idea of Goldstein subdifferential has been used in designing algorithms for nonsmooth $\mathcal{H}_\infty$ control [3, 28, 9, 13]. We will show that such policy search algorithms can be guaranteed to find the global minimum of (6). It is worth mentioning that there are other notions of enlarged subdifferential [2] which can lead to good descent directions for nonsmooth $\mathcal{H}_\infty$ synthesis. In this paper, we focus on the notion of Goldstein subdifferential and related policy search algorithms.

## 3 Optimization Landscape for $\mathcal{H}_\infty$ State-Feedback Control

In this section, we investigate the optimization landscape of the $\mathcal{H}_\infty$ state-feedback policy search problem, and show that any Clarke stationary points of (6) are also global minimum. We start by showing the coerciveness of the $\mathcal{H}_\infty$ objective function (5).

**Lemma 1.** *The $\mathcal{H}_\infty$ objective function $J(K)$ defined by (5) is coercive over the set $\mathcal{K}$ in the sense that for any sequence $\{K^l\}_{l=1}^\infty \subset \mathcal{K}$ we have $J(K^l) \to +\infty$, if either $\|K^l\|_2 \to +\infty$, or $K^l$ converges to an element in the boundary $\partial \mathcal{K}$.*

*Proof.* We will only provide a proof sketch here. A detailed proof is presented in the supplementary material. Suppose we have a sequence $\{K^l\}$ satisfying $\|K^l\|_2 \to +\infty$. We can choose $\mathbf{w} = \{w_0, 0, 0, \cdots\}$ with $\|w_0\| = 1$ and show $J(K^l) \ge w_0^\mathsf{T}(Q + (K^l)^\mathsf{T} R K^l)w_0 \ge \lambda_{\min}(R)\|K^l w_0\|^2$. Clearly, we have used the positive definiteness of $R$ in the above derivation. Then by carefully choosing $w_0$, we can ensure $J(K^l) \to +\infty$ as $\|K^l\|_2 \to +\infty$. Next, we assume $K^l \to K \in \partial \mathcal{K}$. We have $\rho(A - BK) = 1$, and hence there exists some $\omega_0$ such that $(e^{j\omega_0}I - A + BK)$ becomes

singular. Then we can use the positive definiteness of $Q$ to show $J(K^l) \geq \lambda_{\min}^{1/2}(Q)(\|(e^{j\omega_0}I - A + BK^l)^{-1}\| \cdot \|(e^{-j\omega_0}I - A + BK^l)^{-1}\|)^{\frac{1}{2}}$. Notice $\sigma_{\min}(e^{\pm j\omega_0}I - A + BK^l) \to 0$ as $l \to \infty$, which implies $\|(e^{\pm j\omega_0}I - A + BK^l)^{-1}\| \to +\infty$ as $l \to \infty$. Therefore, we have $J(K^l) \to +\infty$ as $K^l \to K \in \partial\mathcal{K}$. More details for the proof can be found in the supplementary material. $\square$

We want to emphasize that the positive definiteness of $(Q, R)$ are crucial for proving the coerciveness of the cost function (5). Built upon Lemma 1, we can obtain the following nice properties of the sublevel sets of (6).

**Lemma 2.** *Consider the $\mathcal{H}_\infty$ state-feedback policy search problem* (6) *with the objective function $J(K)$ defined in* (5). *Under Assumption 1, the sublevel set defined as $\mathcal{K}_\gamma := \{K \in \mathcal{K} : J(K) \leq \gamma\}$ is compact and path-connected for every $\gamma \geq J(K^*)$ where $K^*$ is the global minimum of* (6).

*Proof.* The compactness of $\mathcal{K}_\gamma$ directly follows from the continuity and coerciveness of $J(K)$, and is actually a consequence of [5, Proposition 11.12]. The path-connectedness of the strict sublevel sets for the continuous-time $\mathcal{H}_\infty$ control problem has been proved in [30]. We can slightly modify the proof in [30] to show that the strict sublevel set $\{K \in \mathcal{K} : J(K) < \gamma\}$ is path-connected. Based on the fact that every non-strict sublevel sets are compact, now we can apply [42, Theorem 5.2] to show $\mathcal{K}_\gamma$ is also path-connected. An independent proof based on the non-strict version of the bounded real lemma is also provided in the supplementary material. $\square$

The path-connectedness of $\mathcal{K}_\gamma$ for every $\gamma$ actually implies the uniqueness of the minimizing set in a certain strong sense [42, Sections 2&3]. Due to the space limit, we will defer the discussion on the uniqueness of the minimizing set to the supplementary material. Here, we present a stronger result which is one of the main contributions of our paper.

**Theorem 1.** *Consider the $\mathcal{H}_\infty$ state-feedback policy search problem* (6). *Under Assumption 1, any Clarke stationary point of $J(K)$ is a global minimum.*

A detailed proof is presented in the supplementary material. Here we provide a proof sketch. Since $Q$ and $R$ are positive definite, the non-strict version of the bounded real lemma[2] states that $J(K) \leq \gamma$ if and only if there exists a positive definite matrix $P$ such that the following matrix inequality holds

$$\begin{bmatrix} (A - BK)^\mathsf{T} P (A - BK) - P & (A - BK)^\mathsf{T} P \\ P(A - BK) & P \end{bmatrix} + \begin{bmatrix} Q + K^\mathsf{T} RK & 0 \\ 0 & -\gamma^2 I \end{bmatrix} \preceq 0. \quad (12)$$

The above matrix inequality is linear in $P$ but not linear in $K$. A standard trick from the control theory can be combined with the Schur complement lemma to convert the above matrix inequality condition to another condition which is linear in all the decision variables [6]. Specifically, there exists a matrix function $\mathrm{LMI}(Y, L, \gamma)$ which is linear in $(Y, L, \gamma)$ such that $\mathrm{LMI}(Y, L, \gamma) \preceq 0$ and $Y \succ 0$ if and only if (12) is feasible with $K = LY^{-1}$ and $P = \gamma Y^{-1} \succ 0$. The matrix function $\mathrm{LMI}(Y, L, \gamma)$ involves a larger matrix. Hence we present the analytical formula of $\mathrm{LMI}(Y, L, \gamma)$ in the supplementary material and skip it here. Since $\mathrm{LMI}(Y, L, \gamma)$ is linear in $(Y, L, \gamma)$, we know $\mathrm{LMI}(Y, L, \gamma) \preceq 0$ is just a convex semidefinite programming condition. Based on this convex necessary and sufficient condition for $J(K) \leq \gamma$, we can prove the following important lemma.

**Lemma 3.** *For any $K \in \mathcal{K}$ satisfying $J(K) > J^*$, there exists a matrix direction $d \neq 0$ such that $J'(K, d) \leq J^* - J(K) < 0$, where $J^* = J(K^*)$ and $K^*$ is the global minimum of* (6).

*Proof.* Suppose we have $K = LY^{-1}$ where $(Y, L, J(K))$ is a feasible point for the convex regime $\mathrm{LMI}(Y, L, J(K)) \preceq 0$. In addition, we have $K^* = L^*(Y^*)^{-1}$ where $(Y^*, L^*, J(K^*))$ is a point satisfying $\mathrm{LMI}(Y^*, L^*, J(K^*)) \preceq 0$. Since the LMI condition is convex, the line segment between $(Y, L, J(K))$ and $(Y^*, Q^*, J(K^*))$ is also in this convex set. For any $t > 0$, we know $(Y + t\Delta Y, L + t\Delta L, J(K) + t(J(K^*) - J(K)))$ also satisfies $\mathrm{LMI}(Y + t\Delta Y, L + t\Delta L, J(K) + t(J(K^*) - J(K))) \preceq 0$, where $\Delta L = L^* - L$, and $\Delta Y = Y^* - Y$. Therefore, based on the bounded

---
[2]The difference between the strict and non-strict versions of the bounded real lemma is quite subtle [6, Section 2.7.3]. For completeness, we will provide more explanations for the non-strict version of the bounded real lemma in the supplementary material.

real lemma, we know $J((L + t\Delta L)(Y + t\Delta Y)^{-1}) \leq J(K) + t(J(K^*) - J(K))$. Let's choose $d = \Delta LY^{-1} - LY^{-1}\Delta YY^{-1}$. Then we have

$$J'(K, d) \leq \lim_{t \searrow 0} \left( \frac{J((L + t\Delta L)(Y + t\Delta Y)^{-1}) - J(K)}{t} + o(t) \right) \leq J^* - J(K) < 0.$$

A detailed verification of the above inequality is provided in the supplementary material. Notice $d \neq 0$. If $\Delta LY^{-1} - LY^{-1}\Delta YY^{-1} = 0$, the above argument still works and we reach to the conclusion $J'(K, 0) < 0$. But this is impossible since we always have $J'(K, 0) = 0$. Hence we have $d \neq 0$. This completes the proof for this lemma. $\square$

Now we are ready to provide the proof for Theorem 1. Based on Lemma 3 and the fact that $J(\cdot)$ is subdifferentially regular, the proof can be done by contradiction. Suppose $K^*$ is the global minimum, and $K^\dagger$ is a Clarke stationary point. If $K^\dagger$ is not a global minimum. Then by Lemma 3, there exists $d \neq 0$ such that $J'(K^\dagger, d) < 0$, this contradicts the fact that $J'(K^\dagger, d) \geq 0$ for all $d$ by Corollary 1. Therefore, $K^\dagger$ has to be the global minimum of (6).

The above proof relies on Lemma 3 and the fact that $J$ is subdifferentially regular. Without using the subdifferentially regular property, Lemma 3 itself is not sufficient for proving Theorem 1. It is also worth mentioning that Lemma 3 can be viewed as a modification of the convex parameterization/lifting results in [64, 68] for non-differentiable points.

# 4 Global Convergence of Direct Policy Search on $\mathcal{H}_\infty$ State-Feedback Control

In this section, we first show that Goldstein's subgradient method [25] can be guaranteed to stay in the nonconvex feasible regime $\mathcal{K}$ during the optimization process and eventually converge to the global minimum of (6). The complexity of finding $(\delta, \epsilon)$-stationary points of (6) is also presented. Then we further discuss the convergence guarantees for various implementable forms of Goldstein's subgradient method.

## 4.1 Global convergence and complexity of Goldstein's subgradient Method

We will investigate the global convergence of Goldstein's subgradient method for direct policy search of the optimal $\mathcal{H}_\infty$ state-feedback policy. Goldstein's subgradient method iterates as follows

$$K^{n+1} = K^n - \delta^n F^n / \|F^n\|_2, \tag{13}$$

where $F^n$ is the minimum norm element of the Goldstein subdifferential $\partial_{\delta^n} J(K^n)$. We assume that an initial stabilizing policy is available, i.e. $K^0 \in \mathcal{K}$. The same initial policy assumption has also been made in the global convergence theory for direct policy search on LQR [21]. More recently, some provable guarantees have been obtained for finding such stabilizing policies via direct policy search [52, 51]. Hence such an assumption on the initial policy $K^0$ is reasonable. Our global convergence result relies on the fact that there is a strict separation between any sublevel set of (6) and the boundary of $\mathcal{K}$. This fact is formalized as follows.

**Lemma 4.** *Consider the $\mathcal{H}_\infty$ state-feedback policy search problem* (6) *with the cost function $J(K)$ defined in* (5). *Denote the complement of the feasible set $\mathcal{K}$ as $\mathcal{K}^c$. Suppose Assumption 1 holds and $\gamma \geq J^*$. Then there is a strict separation between the sublevel set $\mathcal{K}_\gamma$ and $\mathcal{K}^c$. In other words, we have $dist(\mathcal{K}_\gamma, \mathcal{K}^c) > 0$.*

*Proof.* Obviously, the set $\mathcal{K}_\gamma \cap \mathcal{K}^c$ is empty (since we know $\mathcal{K}_\gamma \subset \mathcal{K}$). Based on Lemma 2, we know $\mathcal{K}_\gamma$ is compact. Since $\mathcal{K}$ is open, we know $\mathcal{K}^c$ is closed. Therefore, there is a strict separation between $\mathcal{K}_\gamma$ and $\mathcal{K}^c$, and we have $dist(\mathcal{K}_\gamma, \mathcal{K}^c) > 0$. $\square$

Now we are ready to present our main convergence result.

**Theorem 2.** *Consider the $\mathcal{H}_\infty$ state-feedback policy search problem* (6) *with the cost function $J(K)$ defined in* (5). *Suppose Assumption 1 holds, and an initial stabilizing policy is given, i.e. $K^0 \in \mathcal{K}$. Denote $\Delta_0 := \text{dist}(\mathcal{K}_{J(K^0)}, \mathcal{K}^c) > 0$. Choose $\delta^n = \frac{c\Delta_0}{n+1}$ for all $n$ with $c$ being a fixed number in $(0, 1)$. Then Goldstein's subgradient method* (13) *is guaranteed to stay in $\mathcal{K}$ for all $n$. In addition, we have $J(K^n) \to J^*$ as $n \to \infty$.*

*Proof.* We have $\delta^n \leq c\Delta_0 < \Delta_0$ for all $n$. Now we use an induction proof to show $K^n \in \mathcal{K}_{J(K^0)}$ for all $n$. For $n = 0$, we know $K^0 - c\Delta_0 F^0 / \|F^0\|_2$ has to be within the $\Delta_0$ ball around $K^0$ since we know the norm of $F^0 / \|F^0\|_2$ is exactly equal to 1. Since $\Delta_0 := \mathrm{dist}(\mathcal{K}_{J(K^0)}, \mathcal{K}^c) > 0$, we know $K^0 - \delta^0 F^0 / \|F^0\|_2 \in \mathcal{K}$. As a matter of fact, we know $\mathbb{B}_{\delta^0}(K^0)$ has to be a subset of $\mathcal{K}$. Hence we can apply (11) to show that $K^1$ exists and is also in $\mathcal{K}_{J(K^0)}$. Similarly, we can repeat this argument to show $K^n \in \mathcal{K}_{J(K^0)}$ for all $n$. Next, we can apply (11) to every step and then sum the inequalities over all $n$. Then the following inequality holds for all $N$:

$$\sum_{n=0}^{N} \delta_n \|F^n\|_2 \leq J(K^0) - J^* \tag{14}$$

Since we have $\sum_{n=0}^{\infty} \delta^n = +\infty$, we know $\liminf_{n\to\infty} \|F^n\|_2 = 0$. There exists one subsequence $\{i_n\}$ such that $\|F^{i_n}\|_2 \to 0$. For this subsequence, the resultant policy sequence $\{K^{i_n}\}$ is also bounded (notice that the policy parameter sequence stays in the compact set $\mathcal{K}_{J(K^0)}$ for all $n$) and has a convergent subsequence. We can show that the limit of this subsequence is a Clarke stationary point. Hence the function value associated with this subsequence converges to $J^*$. Notice that $J(K^n)$ is monotonically decreasing for the entire sequence $\{n\}$. Hence we have $J(K^n) \to J^*$. $\qquad\square$

We have tried to be brief in giving the above proof. We will present a more detailed proof in the supplementary material. We believe that this is the first result showing that direct policy search can be guaranteed to converge to the global optimal solution of the $\mathcal{H}_\infty$ state-feedback control problem. The above result only provides an asymptotic convergence guarantee to ensure $J(K^n) \to J^*$. One can use a similar argument to establish a finite-time complexity bound for finding the $(\delta, \epsilon)$-stationary points of (6). Such a result is given as follows.

**Theorem 3.** *Consider the $\mathcal{H}_\infty$ problem* (6) *with the cost function* (5). *Suppose Assumption 1 holds, and $K^0 \in \mathcal{K}$. Denote $\Delta_0 := \mathrm{dist}(\mathcal{K}_{J(K^0)}, \mathcal{K}^c) > 0$. For any $\delta < \Delta_0$, we can choose $\delta^n = \delta$ for all $n$ to ensure that Goldstein's subgradient method* (13) *stays in $\mathcal{K}$ and satisfies the following finite-time complexity bound:*

$$\min_{n:0 \leq n \leq N} \|F^n\|_2 \leq \frac{J(K^0) - J^*}{(N+1)\delta} \tag{15}$$

*In other words, we have $\min_{0 \leq n \leq N} \|F^n\|_2 \leq \epsilon$ after $N = \mathcal{O}\left(\frac{\Delta}{\delta\epsilon}\right)$ where $\Delta := J(K^0) - J^*$. For any $\delta < \Delta_0$ and $\epsilon > 0$, the complexity of finding a $(\delta, \epsilon)$-stationary point is $\mathcal{O}\left(\frac{\Delta}{\delta\epsilon}\right)$.*

*Proof.* The above result can be proved using a similar argument from Theorem 2. We can use the same induction argument to show $K^n \in \mathcal{K}_{J(K^0)}$ for all $n$, and (14) holds with $\delta^n = \delta$. Then the desired conclusion directly follows. $\qquad\square$

The complexity for nonsmooth optimization of Lipschitz functions is quite subtle. While the above result gives a reasonable characterization of the finite-time performance of Goldstein's subgradient method on the $\mathcal{H}_\infty$ state-feedback control problem, it does not quantify how fast $J(K^n)$ converges to $J^*$. Recall that $(\delta, \epsilon)$-stationarity means $\mathrm{dist}(0, \partial_\delta J(K)) \leq \epsilon$, while $\epsilon$-stationarity means $\mathrm{dist}(0, \partial_C J(K)) \leq \epsilon$. As commented in [61, 71, 14], $(\delta, \epsilon)$-stationarity does not imply being $\delta$-close to an $\epsilon$-stationary point of $J$. Importantly, the function value of a $(\delta, \epsilon)$-stationary point can be far from $J^*$ even for small $\delta$ and $\epsilon$. Theorem 5 in [71] shows that there is no finite time algorithm that can find $\epsilon$-stationary points provably for all Lipschitz functions. It is still possible that one can develop some finite time bounds for $(J(K^n) - J^*)$ via exploiting other advanced properties of the $\mathcal{H}_\infty$ cost function (5). This is an important future task.

## 4.2 Implementable variants and related convergence results

In practice, it can be difficult to evaluate the minimum norm element of the Goldstein subdifferential. Now we discuss implementable variants of Goldstein's subgradient method and related guarantees.

**Gradient sampling [9, 10, 37].** The gradient sampling (GS) method is the main optimization algorithm used in the robust control package HIFOO [3, 28]. Suppose we can access a first-order

oracle which can evaluate $\nabla J$ for any differentiable points in the feasible set[3]. Based on Rademacher's theorem, a locally Lipschitz function is differentiable almost everywhere. Therefore, for any $K^n \in \mathcal{K}$, we can randomly sample policy parameters over $\mathbb{B}_{\delta^n}(K^n)$ and obtain differentiable points with probability one. For all these sampled differentiable points, the Clarke subdifferential at each point is just the gradient. Then the convex hull of these sampled gradients can be used as an approximation for the Goldstein subdifferential $\partial_{\delta^n} K^n$. The minimum norm element from the convex hull of the sampled gradients can be solved via a simple convex quadratic program, and is sufficient for generating a reasonably good descent direction for updating $K^{n+1}$ as long as we sample at least $(n_x n_u + 1)$ differentiable points for each $n$ [9]. In the unconstrained setup, the cluster points of the GS algorithm can be guaranteed to be Clarke stationary [37, 9]. Such a result can be combined with Theorem 1 and Lemma 4 to show the global convergence of the GS method on the $\mathcal{H}_\infty$ state-feedback synthesis problem. The following theorem will be treated formally in the supplementary material.

**Theorem 4** (Informal statement). *Consider the policy optimization problem* (6) *with the $\mathcal{H}_\infty$ cost function defined in* (5). *Suppose Assumption 1 holds, and $K^0 \in \mathcal{K}$. The iterations generated from the trust-region version of the GS method (described in [37, Section 4.2] and restated in the supplementary material) can be guaranteed to stay in $\mathcal{K}$ for all iterations and achieve $J(K^n) \to J^*$ with probability one.*

**Non-derivative sampling (NS) [38].** The NS method can be viewed as the derivative-free version of the GS algorithm. Suppose we only have the zeroth-order oracle which can evaluate the function value $J(K)$ for $K \in \mathcal{K}$. The main difference between NS and GS is that the NS algorithm relies on estimating the gradient from function values via Gupal's estimation method. In the unconstrained setting, the cluster points of the NS method can be guaranteed to be Clarke stationary with probability one [38, Theorem 3.8]. We can combine [38, Theorem 3.8] with our results (Theorem 1 and Lemma 4) to prove the global convergence of NS in our setting. A detailed discussion is given in the supplementary material.

**Model-free implementation of NS.** When the system model is unknown, there are various methods available for estimating the $\mathcal{H}_\infty$-norm from data [45, 46, 57, 54, 69, 50, 67, 66]. Based on our own experiences/tests, the multi-input multi-output (MIMO) power iteration method [49] works quite well as a stochastic zeroth-order oracle for the purpose of implementing NS in the model-free setting. While the sample complexity for model-free NS is unknown, we will provide some numerical justifications to show that such a model-free implementation closely tracks the convergence behaviors of its model-based counterpart.

**Interpolated normalized gradient descent (INGD) with finite-time complexity.** No finite-time guarantees for finding $(\delta, \epsilon)$-stationary points have been reported for the GS/NS methods. In [71, 14], the INGD method has been developed as another implementable variant of Goldstein's subgradient method, and is proved to satisfy high-probability finite-time complexity bounds for finding $(\delta, \epsilon)$-stationary points of Lipschitz functions. INGD uses an iterative sampling strategy to generate a descent direction which serves a role similar to the minimal norm element of the Goldstein subdifferential. A first-order oracle for differentiable points is needed for implementing the version of INGD in [14]. It has been show [71, 14] that for unconstrained nonsmooth optimization of $L$-Lipschitz functions[4], the INGD algorithm can be guaranteed to find the $(\delta, \epsilon)$-stationary point with the high-probability iteration complexity $\mathcal{O}\left(\frac{\Delta L^2}{\epsilon^3 \delta} \log(\frac{\Delta}{p \delta \epsilon})\right)$, where $\Delta := J(K^0) - J^*$ is the initial function value gap, and $p$ is the failure probability (i.e. the optimization succeeds with the probability $(1 - p)$). We can combine the proofs for [14, Theorem 2.6] and Theorem 3 to obtain the following complexity result for our $\mathcal{H}_\infty$ setting. A formal treatment is given in the supplementary material.

**Theorem 5** (Informal statement). *Consider the policy optimization problem* (6) *with the $\mathcal{H}_\infty$ cost function defined in* (5). *Suppose Assumption 1 holds, and the initial policy is stabilizing, i.e. $K^0 \in \mathcal{K}$. Denote $\Delta_0 := \mathrm{dist}(\mathcal{K}_{J(K^0)}, \mathcal{K}^c) > 0$, and let $L_0$ be the Lipschitz constant of $J(K)$ over the set $\mathcal{K}_{J(K^0)}$. For any $\delta < \Delta_0$, we can choose $\delta^n = \delta$ for all $n$ to ensure that the iterations of the INGD algorithm stay in $\mathcal{K}$ almost surely, and find a $(\delta, \epsilon)$-stationary point with the high-probability iteration complexity $\mathcal{O}\left(\frac{\Delta L_0^2}{\epsilon^3 \delta} \log(\frac{\Delta}{p \delta \epsilon})\right)$, where $p$ is the failure probability.*

---

[3] When $(A, B)$ is known, one can calculate the $\mathcal{H}_\infty$ gradient at differential points using the chain rule in [2]. More explanations can be found in the supplementary material.

[4] We slightly abuse our notation by denoting the Lipschitz constant as $L$. Previously, we have used $L$ to denote a particular matrix used in the LMI formulation for $\mathcal{H}_\infty$ state-feedback synthesis.

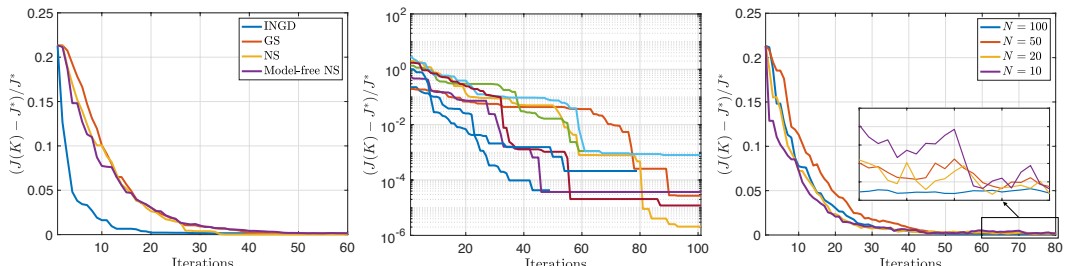

Figure 1: Simulation results. Left: The trajectory of relative error of GS, NS, INGD, and Model-free NS methods on (16). Middle: The trajectory of relative optimality gap of 8 randomly generated cases for NS method. Right: The trajectory of Model-free NS method with more noisy oracle on (16).

## 5 Numerical Simulations

To support our theory, we provide some numerical simulations in this section. The left plot in Figure 1 shows that GS, NS, INGD, and model-free NS work well for the following example:

$$A = \begin{bmatrix} 1 & 0 & -5 \\ -1 & 1 & 0 \\ 0 & 0 & 1 \end{bmatrix}, \ B = \begin{bmatrix} 1 \\ 0 \\ -1 \end{bmatrix}, \ Q = \begin{bmatrix} 2 & -1 & 0 \\ -1 & 2 & -1 \\ 0 & -1 & 2 \end{bmatrix}, \ R = 1. \quad (16)$$

For this example, we have $J^* = 7.3475$. We initialize from $K^0 = [0.4931 \quad -0.1368 \quad -2.2654]$, which satisfies $\rho(A - BK^0) = 0.5756 < 1$. The hyperparameter choices are detailed in the supplementary material. We can see that model-free NS closely tracks the trajectory of NS and works well. In the middle plot of Figure 1, we test the NS method on randomly generated cases. We set $A \in \mathbb{R}^{3\times3}$ to be $I + \xi$, where each element of $\xi \in \mathbb{R}^{3\times3}$ is sampled uniformly from $[0, 1]$. For $B \in \mathbb{R}^{3\times1}$, each element is uniformly sampled from $[0, 1]$. We have $Q = I + \zeta I \in \mathbb{R}^{3\times3}$ with $\zeta$ uniformly sampled from $[0, 0.1]$, and $R \in \mathbb{R}$ uniformly sampled from $[1, 1.5]$. For each experiment, the initial condition $K^0 \in \mathbb{R}^{1\times3}$ is also randomly sampled such that $\rho(A - BK^0) < 1$. The NS method converges globally for all the cases. In the right plot, we focus on the model-free setting for (16). We decrease the number of samples used in the $\mathcal{H}_\infty$ estimation and show how this increases the noise in the zeroth-order $\mathcal{H}_\infty$ oracle and worsens the convergence behaviors of the model-free NS method. Nevertheless, the model-free NS method tracks its model-based counterpart with enough samples. More numerical results can be found in the supplementary material.

## 6 Conclusions and Future Work

In this paper, we developed the global convergence theory for direct policy search on the $\mathcal{H}_\infty$ state-feedback synthesis problem. Although the resultant policy optimization formulation is nonconvex and nonsmooth, we managed to show that any Clarke stationary points for this problem are actually global minimum, and the concept of Golstein subdifferential can be used to build direct policy search algorithms which are guaranteed to converge to the global optimal solutions. The finite-time guarantees in this paper are developed only for finding $(\delta, \epsilon)$-stationary points. An important future task is to investigate the finite-time bounds for the optimality gap (i.e. $J(K^n) - J^*$) as well as the sample complexity of direct policy search on model-free $\mathcal{H}_\infty$ control. It is also of great interests to investigate the convergence properties of direct policy search in nonlinear/output-feedback settings[5].

## Acknowledgments and Disclosure of Funding

This work is generously supported by the NSF award CAREER-2048168 and the 2020 Amazon research award. The authors would like to thank Michael L. Overton, Maryam Fazel, Yang Zheng, Peter Seiler, Geir Dullerud, Aaron Havens, Darioush Kevian, Kaiqing Zhang, Na Li, Mehran Mesbahi, Tamer Başar, Mihailo Jovanovic, and Javad Lavaei for the valuable discussions, as well as the helpful suggestions from the anonymous reviewers of NeurIPS.

---

[5]Some discussions on possible extensions along this direction have been given in the supplementary material.

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
