# Supplementary Material

## A  More Details on Problem Formulation and Background

### A.1  Difference between $\mathcal{H}_\infty$ control and mixed $\mathcal{H}_2/\mathcal{H}_\infty$ design

In this paper, we are interested in $\mathcal{H}_\infty$ control whose objective is to design a state-feedback policy $K$ that stabilizes the closed-loop system and minimizes the closed-loop $\mathcal{H}_\infty$ norm (or equivalently the $\ell_2 \to \ell_2$ induced norm) of $G_K$ at the same time. As discussed in the main paper, $\mathcal{H}_\infty$ control can be formulated as $\min_{K\in\mathcal{K}}\|G_K\|_{2\to2}$. We have mentioned that such a formulation corresponds to a worst-case assumption on the disturbance $\mathbf{w}$. Since $\|G_K\|_{2\to2}$ is a nonsmooth function in $K$, the $\mathcal{H}_\infty$ control leads to a nonsmooth optimization problem. The global optimal $\mathcal{H}_\infty$ norm for this problem is denoted as $\gamma^*$.

In contrast, the design objective of the mixed $\mathcal{H}_2/\mathcal{H}_\infty$ control is to synthesize a a linear state-feedback policy which minimizes an upper bound on the $\mathcal{H}_2$ cost and satisfies an additional $\mathcal{H}_\infty$-robustness requirement in the form of $\|G_K\|_{2\to2} < \gamma$ with some pre-specified value of $\gamma$. Recall that we denote the non-strict $\gamma$-admissible sublevel set of the $\mathcal{H}_\infty$ objective function as $\mathcal{K}_\gamma = \{K \in \mathcal{K} : \|G_K\|_{2\to2} \le \gamma\}$. Similarly, we definie the strict $\gamma$-admissible sublevel set of the $\mathcal{H}_\infty$ objective function as $\tilde{\mathcal{K}}_\gamma = \{K \in \mathcal{K} : \|G_K\|_{2\to2} < \gamma\}$. Then for any $\gamma > \gamma^*$, the $\mathcal{H}_2/\mathcal{H}_\infty$ mixed design problem can be formulated as

$$\min_{K\in\tilde{\mathcal{K}}_\gamma} \operatorname{tr}(P_K)$$

where $P_K$ is the minimal positive definite solution to the following Riccati equation

$$(A - BK)^\mathsf{T}(P_K + P_K(\gamma^2 I - P_K)^{-1}P_K)(A - BK) + Q + K^\mathsf{T}RK - P_K = 0.$$

Notice that as $\gamma \to \infty$, the above mixed design problem reduces to the LQR problem. For finite $\gamma$, $\operatorname{tr}(P_K)$ provides an upper bound for the LQR cost. For the mixed $\mathcal{H}_2/\mathcal{H}_\infty$ control problem, the closed-loop $\mathcal{H}_\infty$ norm $\|G_K\|_{2\to2}$ only appears in the constraint. One can show that cost function $\operatorname{tr}(P_K)$ is still differentiable in the policy space for any $K \in \tilde{\mathcal{K}}$. The main result in [73] states that given an initial policy $K^0 \in \tilde{\mathcal{K}}_\gamma$ for $\gamma > \gamma^*$, then the natural policy gradient method with well-chosen stepsize is guaranteed to stay in $\tilde{\mathcal{K}}_\gamma$ and fine the optimal policy minimizes $\operatorname{tr}(P_K)$. This means that the natural policy gradient method applied to this mixed design problem can improve the average performance (via minimizing $\operatorname{tr}(P_K)$) while maintaining the robustness level $\gamma$. A missing step is how to use direct policy search to obtain such an initial policy satisfying $\|G_{K^0}\|_{2\to2} < \gamma$ in the first place. One by-product of the results in this paper is a direct policy search method for obtaining such initial policy provably.

It is worth mentioning that it is also possible to to tackle the $\mathcal{H}_\infty$ state-feedback synthesis via solving a sequence of mixed $\mathcal{H}_2/\mathcal{H}_\infty$ design problems, and the approximate central path algorithm in [35] is developed based on such an idea. However, there is no global convergence theory reported for such successive minimization approaches [35].

### A.2  Formulating $\mathcal{H}_\infty$ control with arbitrary $\ell_2$ bound on the disturbance

To address the worst-case disturbance $\mathbf{w}$, one may be interested in the following problem formulation

$$\min_{\mathbf{u}} \max_{\mathbf{w}:\|\mathbf{w}\|\le\Lambda} \sum_{t=0}^{\infty}(x_t^\mathsf{T}Qx_t + u_t^\mathsf{T}Ru_t) \tag{A.1}$$

where $\Lambda$ is an arbitrary positive number. An interesting fact is that (A.1) still leads to the same policy optimization problem $\min_{K\in\mathcal{K}}\|G_K\|_{2\to2}$ regardless of the value of $\Lambda$. Now we briefly explain this fact. Again, it is well-know that it suffices to consider linear state-feedback policies for solving (A.1). Notice that $G_K$ is a linear operator for any fixed $K$. Therefore, it is straightforward to verify

$$\max_{\mathbf{w}:\|\mathbf{w}\|\le\Lambda} \sum_{t=0}^{\infty}(x_t^\mathsf{T}Qx_t + u_t^\mathsf{T}Ru_t) = \Lambda^2\|G_K\|_{2\to2}^2$$

Consequently, (A.1) can be equivalently formulated as $\min_{K\in\mathcal{K}}\Lambda^2\|G\|_{2\to2}^2$. The constant $\Lambda^2$ can be removed without changing the optimization problem. Therefore, no matter what positive value we use for $\Lambda$, (A.1) is always equivalent to $\min_{K\in\mathcal{K}}\|G\|_{2\to2}$.

## A.3 More explanations for Proposition 1

Now we provide some explanations for Proposition 1 which states several facts observed in the existing literature. The continuity of the cost function (5) can be seen from its analytic form. The openness of $\mathcal{K}$ can be seen from the fact that the spectral radius $\rho(A - BK)$ is a continuous function of $K$. The unboundedness of $\mathcal{K}$ has been proved in [8]. To see that $\mathcal{K}$ can be nonconvex, we provide a simple example here. Suppose $A = B = I$, and we choose two policies as follows

$$K = \begin{bmatrix} 1 & 0 \\ -1 & 1 \end{bmatrix}, \quad K' = \begin{bmatrix} 1 & -5 \\ 0 & 1 \end{bmatrix}.$$

Then one can easily check that $K, K' \in \mathcal{K}$, while their convex combination $\frac{1}{2}(K_1 + K_2) \notin \mathcal{K}$.

Finally, we provide an example to show $J(K)$ is nonconvex in $K$. Suppose $A = B = Q = R = I$ and choose two policies as follows:

$$K = \begin{bmatrix} 0.7 & -3.4 \\ 0.2 & 1 \end{bmatrix}, \quad K' = \begin{bmatrix} 0 & -1.3 \\ 0.6 & 1.2 \end{bmatrix}.$$

Let $K'' = \frac{1}{2}(K + K')$, then we have $J(K'') = 35.99$ and $\frac{1}{2}(J(K) + J(K')) = 13.3$. Hence $J(K'') > 0.5J(K) + 0.5J(K')$, which shows $J(K)$ is nonconvex in $K$.

## A.4 Proof of Proposition 2

Proposition 2 is a consequence of the chain rule stated in [11, Theorem 2.3.10]. As mentioned previously, Proposition 2 should be considered as a well-known fact, and we do not claim any credits in proving it. Although not explicitly stated, the proof of Proposition 2 has been hinted in the discussion of [2, Section III]. We only present the proof for completeness.

To apply [11, Theorem 2.3.10], we can rewrite the closed-loop $\mathcal{H}_\infty$ norm $\|G_K\|_{2\to2}$ as a composition $g_2 \circ g_1(K)$ where $g_1$ maps the policy parameter $K$ to a stable transfer function $(Q + K^\mathsf{T} RK)^{\frac{1}{2}}(zI - A + BK)^{-1}$ which lives in the infinite-dimensional $\mathcal{H}_\infty$ Hardy space, and $g_2$ maps any stable transfer function in that Hardy space to its $\mathcal{H}_\infty$ norm. It is well known that the infinite-dimensional $\mathcal{H}_\infty$ Hardy space consisting of all stable LTI systems is a Banach space [78]. We also know that $g_2$ is convex on the Hardy space (by triangle inequality of the $\mathcal{H}_\infty$ norm). The convexity of $g_2$ implies that $g_2$ is also subdifferentially regular. One can also show that $g_1$ is a strictly differentiable mapping from $K$ to the Hardy space given $\rho(A - BK) < 1$ and $Q$ being positive definite. Now we can immediately apply [11, Theorem 2.3.10] to obtain the desired conclusion in Proposition 2.

**Remark 1.** *For readability, we also provide some brief explanations for the fact that the mapping $g_1$ is strictly differentiable. For each $K \in \mathcal{K}$, $g_1$ maps $K$ to an infinite-dimensional bounded operator which maps from $\ell_2^{n_w}$ to $\ell_2^{n_x}$ and has a Toeplitz structure. The Toeplitz structure can be combined with Fubini's theorem to show simple upper bounds for the operator norm of $g_1$. One such upper bound is provided by $\|(Q + K^\mathsf{T} RK)^{\frac{1}{2}}\| \sum_{k=0}^{\infty} \|(A - BK)^k\|$, which is obviously finite for $K \in \mathcal{K}$. Such a bound allows us to study the operator $g_1$ via treating it as an infinite-dimensional Toeplitz matrix whose properties are completely determined by its first block column. By definition, $g_1$ admits a strict derivative at $K$ if the following operator norm convergence result holds for any $V$:*

$$\lim_{K' \to K, \, t \downarrow 0} \|(g_1(K' + tV) - g_1(K'))/t - \langle D_s g_1(K), V \rangle\| = 0, \tag{A.2}$$

*where $\|\cdot\|$ denotes the operator norm, and $D_s g_1$ is the candidate for the strict derivative defined via taking derivatives of the infinite-dimensional matrix $g_1$ in a block-by-block manner. Since we know $\rho(A - BK) < 1$, the Toeplitz structure of $g_1$ can be combined with Fubini's theorem again to prove the existence of $D_s g_1$ and provide simple upper bounds for the operator norm on the left side of (A.2), leading to the desired operator norm convergence result in (A.2).*

## A.5 A non-strict version of the Bounded Real Lemma

The bounded real lemma (also referred to as the Kalman–Yakubovich–Popov (KYP) lemma) is an important tool for characterizing the sublevel sets of the closed-loop $\mathcal{H}_\infty$ norm. There is some subtlety in the assumptions needed for different versions of the bounded real lemma. We briefly

clarify this subtlety here. Consider the following closed-loop linear system:

$$x_{t+1} = A_{\mathrm{cl}}x_t + B_{\mathrm{cl}}w_t$$
$$z_t = C_{\mathrm{cl}}x_t, \tag{A.3}$$

where $\rho(A_{\mathrm{cl}}) < 1$. Recall that the closed-loop $\mathcal{H}_\infty$ norm of the above system (from $\{w_t\}$ to $\{z_t\}$) is defined as

$$H_{cl} = \sup_{\omega \in [0,2\pi]} \lambda_{\max}^{1/2}\big(B_{\mathrm{cl}}^{\mathsf{T}}(e^{-j\omega}I - A_{\mathrm{cl}})^{-\mathsf{T}}C_{\mathrm{cl}}^{\mathsf{T}}C_{\mathrm{cl}}(e^{j\omega}I - A_{\mathrm{cl}})^{-1}B_{\mathrm{cl}}\big).$$

The strict version of the bounded real lemma [78, Theorem 21.12] states that $H_{\mathrm{cl}} < \gamma$ if and only if there exists a positive definite matrix $P$ such that

$$\begin{bmatrix} A_{\mathrm{cl}}^{\mathsf{T}}PA_{\mathrm{cl}} - P & A_{\mathrm{cl}}^{\mathsf{T}}PB_{\mathrm{cl}} \\ B_{\mathrm{cl}}^{\mathsf{T}}PA_{\mathrm{cl}} & B_{\mathrm{cl}}^{\mathsf{T}}PB_{\mathrm{cl}} \end{bmatrix} + \begin{bmatrix} C_{\mathrm{cl}}^{\mathsf{T}}C_{\mathrm{cl}} & 0 \\ 0 & -\gamma^2 I \end{bmatrix} \prec 0.$$

The above strict-version version of the bounded real lemma does not require any assumptions on $(A_{\mathrm{cl}}, B_{\mathrm{cl}}, C_{\mathrm{cl}})$. There is also a non-strict version of the bounded real lemma which requires the extra assumption that $(A_{\mathrm{cl}}, B_{\mathrm{cl}}, C_{\mathrm{cl}})$ is minimal (i.e. $(A_{\mathrm{cl}}, B_{\mathrm{cl}})$ is controllable and $(A_{\mathrm{cl}}, C_{\mathrm{cl}})$ is observable). Consider (A.3) with $(A_{\mathrm{cl}}, B_{\mathrm{cl}}, C_{\mathrm{cl}})$ being a minimal realization. Then the non-strict version of the bounded real lemma[6] states that $H_{\mathrm{cl}} \leq \gamma$ if and only if there exists a positive definite matrix $P$ such that

$$\begin{bmatrix} A_{\mathrm{cl}}^{\mathsf{T}}PA_{\mathrm{cl}} - P & A_{\mathrm{cl}}^{\mathsf{T}}PB_{\mathrm{cl}} \\ B_{\mathrm{cl}}^{\mathsf{T}}PA_{\mathrm{cl}} & B_{\mathrm{cl}}^{\mathsf{T}}PB_{\mathrm{cl}} \end{bmatrix} + \begin{bmatrix} C_{\mathrm{cl}}^{\mathsf{T}}C_{\mathrm{cl}} & 0 \\ 0 & -\gamma^2 I \end{bmatrix} \preceq 0.$$

For the main result in this paper, the proof require the above non-strict version of the bounded real lemma. Specifically, in our setting, we have $\|G_K\|_{2\to 2} = H_{\mathrm{cl}}$ if we choose $A_{\mathrm{cl}} = A - BK$, $B_{\mathrm{cl}} = I$, and $C_{\mathrm{cl}} = (Q + K^{\mathsf{T}}RK)^{\frac{1}{2}}$. Since $Q$ is positive definite, such choice of $(A_{\mathrm{cl}}, B_{\mathrm{cl}}, C_{\mathrm{cl}})$ leads to a minimal realization, and hence it is valid to apply the non-strict version of the bounded real lemma to characterize the non-strict sublevel sets of $\|G_K\|_{2\to 2}$.

## B  Detailed Proofs of Our Main Results in Sections 3 & 4

In the main paper, we have only provided the proof sketches for Lemma 1, Lemma 2, Theorem 1, Lemma 3, and Theorem 2. In this section, we provide detailed proofs for these results.

### B.1  Proof for Lemma 1

Suppose we have a sequence $\{K^l\}$ satisfying $\|K^l\|_2 \to +\infty$. We can choose $\mathbf{w} = \{w_0, 0, 0, \cdots\}$ with $\|w_0\| = 1$. Then we have:

$$\begin{aligned} J(K^l) &= \max_{\mathbf{w}:\|\mathbf{w}\|\leq 1} \sum_{t=0}^{\infty} x_t^{\mathsf{T}}(Q + (K^l)^{\mathsf{T}}RK^l)x_t \\ &\geq w_0^{\mathsf{T}}(Q + (K^l)^{\mathsf{T}}RK^l)w_0 \\ &= w_0^{\mathsf{T}}Qw_0 + (K^lw_0)^{\mathsf{T}}R(K^lw_0) \\ &\geq \lambda_{\min}(R)\|K^lw_0\|^2, \end{aligned}$$

where the first inequality holds since we plugged into a specific $\mathbf{w}$ over the max operation and the matrix $Q + (K^l)^{\mathsf{T}}RK^l$ is positive definite. The second inequality uses the fact that $R \geq \lambda_{\min}(R)I$. Then by carefully choosing $w_0$, we can ensure $J(K^l) \to +\infty$ as $\|K^l\| \to +\infty$.

Next, we assume $K^l \to K$ where $K$ is on the boundary $\partial\mathcal{K}$. Clearly we have $\rho(A - BK) = 1$. We will use a frequency-domain argument to prove $J(K^l) \to +\infty$. Since $\rho(A - BK) = 1$, there exists some $\omega_0$ such that the matrix $(e^{j\omega_0}I - A + BK)$ becomes singular. Obviously, for the same $\omega_0$, the

---
[6]See [6, Section 2.7.3] for the continuous-time counterpart.

matrix $(e^{-j\omega_0}I - A + BK)$ is also singular. Therefore, we have:

$$
\begin{aligned}
J(K^l) &= \sup_{\omega \in [0,2\pi]} \lambda_{\max}^{1/2}\big((e^{-j\omega}I - A + BK^l)^{-\mathsf{T}}(Q + (K^l)^{\mathsf{T}}RK^l)(e^{j\omega}I - A + BK^l)^{-1}\big) \\
&= \sup_{\omega \in [0,2\pi]} \|(e^{-j\omega}I - A + BK^l)^{-\mathsf{T}}(Q + (K^l)^{\mathsf{T}}RK^l)(e^{j\omega}I - A + BK^l)^{-1}\|^{1/2} \\
&\geq \lambda_{\min}^{1/2}(Q) \sup_{\omega \in [0,2\pi]} (\|(e^{-j\omega}I - A + BK^l)^{-1}\| \cdot \|(e^{j\omega}I - A + BK^l)^{-1}\|)^{1/2} \\
&\geq \lambda_{\min}^{1/2}(Q)(\|(e^{-j\omega_0}I - A + BK^l)^{-1}\| \cdot \|(e^{j\omega_0}I - A + BK^l)^{-1}\|)^{1/2}.
\end{aligned}
$$

Clearly, the above argument relies on the fact that $Q$ is positive definite. Notice that we have $\rho(A - BK^l) < 1$ for each $l$, and hence we have $\sigma_{\min}(e^{\pm j\omega_0}I - A + BK^l) > 0$, i.e. the smallest singular values of $(e^{\pm j\omega_0}I - A + BK^l)$ are strictly positive for all $l$. As $l \to \infty$, one can show that the matrices $(e^{\pm j\omega_0}I - A + BK^l)$ converge to the singular matrices $(e^{\pm j\omega_0}I - A + BK)$ with $K = \lim_{l\to\infty} K^l \in \partial\mathcal{K}$. Hence we have $\sigma_{\min}(e^{\pm j\omega_0}I - A + BK^l) \to 0$ as $l \to \infty$, which implies $\|(e^{\pm j\omega_0}I - A + BK^l)^{-1}\| \to +\infty$ as $l \to \infty$. Therefore, we have $J(K^l) \to +\infty$ as $K^l \to K \in \partial\mathcal{K}$. This completes the proof.

## B.2 Proof for Lemma 2

We first show the compactness of $\mathcal{K}_\gamma$. Since $J$ is continuous, we know $\mathcal{K}_\gamma = \{K \in \mathcal{K} : J(K) \leq \gamma\}$ is a closed set. It remains to show $\mathcal{K}_\gamma$ is bounded. Suppose there exist $\gamma > 0$ such that $\mathcal{K}_\gamma$ is unbounded. Then there exists a sequence $\{K^l\}_{l=1}^\infty \subset \mathcal{K}_\gamma$ such that $\|K^l\|_2 \to +\infty$ as $l \to \infty$. But by coerciveness of $J(K)$, we must have $J(K^l) \to +\infty$ as well, which contradicts that $J(K^l) \leq \gamma$ for all $l$. Hence $\mathcal{K}_\gamma$ is bounded. Therefore, $\mathcal{K}_\gamma$ is compact. The path-connectedness of the strict sublevel sets for the continuous-time $\mathcal{H}_\infty$ control problem has been proved in [30]. We can slightly modify the proof in [30] to show that the strict sublevel set $\{K \in \mathcal{K} : J(K) < \gamma\}$ is path-connected. Based on the fact that every non-strict sublevel sets are compact, we can apply [42, Theorem 5.2] to show $\mathcal{K}_\gamma$ is also path-connected.

We can also prove the path-connectedness of $\mathcal{K}_\gamma$ by using the non-strict version of the bounded real lemma reviewed in Section A.5. This proof is more self-contained, and hence also included here. Since $Q$ and $R$ are positive definite, the non-strict version of the bounded real lemma [6, Section 2.7.3] states that $J(K) \leq \gamma$ if and only if there exists a positive definite matrix $P$ such that the following non-strict matrix inequality holds

$$
\begin{bmatrix} (A - BK)^{\mathsf{T}}P(A - BK) - P & (A - BK)^{\mathsf{T}}P \\ P(A - BK) & P \end{bmatrix} + \begin{bmatrix} Q + K^{\mathsf{T}}RK & 0 \\ 0 & -\gamma^2 I \end{bmatrix} \preceq 0. \tag{B.1}
$$

The above matrix inequality is linear in $P$ but not linear in $K$. A standard trick from the control theory can be combined with the Schur complement lemma to convert the above matrix inequality condition to another condition which is linear in all the decision variable [6]. Specifically, there exists a matrix function $\mathrm{LMI}(Y, L, \gamma)$ which is linear in $(Y, L, \gamma)$ such that $Y \succ 0$ and $\mathrm{LMI}(Y, L, \gamma) \preceq 0$ if and only if (B.1) is feasible with $K = LY^{-1}$, and $P = \gamma Y^{-1}$. For completeness, we provide the detailed derivation of $\mathrm{LMI}(Y, L, \gamma)$ as follows.

**Step 1**: Let $\tilde{P} = \frac{1}{\gamma}P$, dividing both sides of (B.1) by $\gamma$. Then by Schur complement lemma, (B.1) can be rewritten as:

$$
\begin{bmatrix} (A - BK)^{\mathsf{T}}\tilde{P}(A - BK) - \tilde{P} + \frac{1}{\gamma}K^{\mathsf{T}}RK & (A - BK)^{\mathsf{T}}\tilde{P} & I \\ \tilde{P}(A - BK) & \tilde{P} - \gamma I & 0 \\ I & 0 & -\gamma Q^{-1} \end{bmatrix} \preceq 0, \tag{B.2}
$$

To see this, noticing that $-\gamma Q^{-1}$ is negative definite and (B.1) can be rewritten as:

$$
\begin{bmatrix} (A - BK)^{\mathsf{T}}\tilde{P}(A - BK) - \tilde{P} + \frac{1}{\gamma}K^{\mathsf{T}}RK & (A - BK)^{\mathsf{T}}\tilde{P} \\ \tilde{P}(A - BK) & \tilde{P} - \gamma I \end{bmatrix} - \begin{bmatrix} I \\ 0 \end{bmatrix} (-\frac{1}{\gamma}Q) \begin{bmatrix} I & 0 \end{bmatrix} \preceq 0. \tag{B.3}
$$

**Step 2**: Now we can apply Schur complement lemma to (B.2) again to show its equivalence to:

$$\begin{bmatrix} -\tilde{P} + \frac{1}{\gamma}K^{\mathsf{T}}RK & 0 & I & (A-BK)^{\mathsf{T}}\tilde{P} \\ 0 & -\gamma I & 0 & \tilde{P} \\ I & 0 & -\gamma Q^{-1} & 0 \\ \tilde{P}(A-BK) & \tilde{P} & 0 & -\tilde{P} \end{bmatrix} \preceq 0. \tag{B.4}$$

To see this, noticing that $-\tilde{P}$ is negative definite and the LHS of (B.2) can be rewritten as:

$$\begin{bmatrix} -\tilde{P} + \frac{1}{\gamma}K^{\mathsf{T}}RK & 0 & I \\ 0 & -\gamma I & 0 \\ I & 0 & -\gamma Q^{-1} \end{bmatrix} - \begin{bmatrix} (A-BK)^{\mathsf{T}}\tilde{P} \\ \tilde{P} \\ 0 \end{bmatrix} (-\tilde{P}^{-1}) \begin{bmatrix} \tilde{P}(A-BK) & \tilde{P} & 0 \end{bmatrix}. \tag{B.5}$$

**Step 3**: Then we left and right multiply $\mathrm{diag}(\begin{bmatrix} \tilde{P}^{-1} & I & I & \tilde{P}^{-1} \end{bmatrix})$ on both sides of (B.4), this will not change the definiteness of (B.4) and leads the LHS of (B.4) to:

$$\begin{bmatrix} \tilde{P}^{-1} & 0 & 0 & 0 \\ 0 & I & 0 & 0 \\ 0 & 0 & I & 0 \\ 0 & 0 & 0 & \tilde{P}^{-1} \end{bmatrix} \begin{bmatrix} -\tilde{P} + \frac{1}{\gamma}K^{\mathsf{T}}RK & 0 & I & (A-BK)^{\mathsf{T}}\tilde{P} \\ 0 & -\gamma I & 0 & \tilde{P} \\ I & 0 & -\gamma Q^{-1} & 0 \\ \tilde{P}(A-BK) & \tilde{P} & 0 & -\tilde{P} \end{bmatrix} \begin{bmatrix} \tilde{P}^{-1} & 0 & 0 & 0 \\ 0 & I & 0 & 0 \\ 0 & 0 & I & 0 \\ 0 & 0 & 0 & \tilde{P}^{-1} \end{bmatrix} \tag{B.6}$$

$$= \begin{bmatrix} -\tilde{P}^{-1} + \frac{1}{\gamma}\tilde{P}^{-1}K^{\mathsf{T}}RK\tilde{P}^{-1} & 0 & \tilde{P}^{-1} & \tilde{P}^{-1}(A-BK)^{\mathsf{T}} \\ 0 & -\gamma I & 0 & I \\ \tilde{P}^{-1} & 0 & -\gamma Q^{-1} & 0 \\ (A-BK)\tilde{P}^{-1} & I & 0 & -\tilde{P}^{-1} \end{bmatrix} \tag{B.7}$$

**Step 4**: Substituting $Y = \tilde{P}^{-1}$ and $KY = L$ into the above matrix leads (B.4) to:

$$\begin{bmatrix} -Y + \frac{1}{\gamma}L^{\mathsf{T}}RL & 0 & Y & (AY-BL)^{\mathsf{T}} \\ 0 & -\gamma I & 0 & I \\ Y & 0 & -\gamma Q^{-1} & 0 \\ AY-BL & I & 0 & -Y \end{bmatrix} \preceq 0 \tag{B.8}$$

**Step 5**: Furthermore, (B.8) is equivalent to:

$$\begin{bmatrix} -Y & 0 & Y & (AY-BL)^{\mathsf{T}} \\ 0 & -\gamma I & 0 & I \\ Y & 0 & -\gamma Q^{-1} & 0 \\ AY-BL & I & 0 & -Y \end{bmatrix} - \begin{bmatrix} L^{\mathsf{T}} \\ 0 \\ 0 \\ 0 \end{bmatrix} (-\frac{1}{\gamma}R) \begin{bmatrix} L & 0 & 0 & 0 \end{bmatrix} \preceq 0 \tag{B.9}$$

Applying Schur complement lemma to (B.9) again leads to:

$$\mathrm{LMI}(Y,L,\gamma) := \begin{bmatrix} -Y & 0 & Y & (AY-BL)^{\mathsf{T}} & L^{\mathsf{T}} \\ 0 & -\gamma I & 0 & I & 0 \\ Y & 0 & -\gamma Q^{-1} & 0 & 0 \\ AY-BL & I & 0 & -Y & 0 \\ L & 0 & 0 & 0 & -\gamma R^{-1} \end{bmatrix} \preceq 0. \tag{B.10}$$

This proves the equivalence between (B.10) and (B.1). Therefore, we have:

$$\begin{aligned} & \{K \in \mathcal{K} : J(K) \leq \gamma\} \\ \Longleftrightarrow & \{K : \text{(B.1) is feasible, } P \succ 0\} \\ \Longleftrightarrow & \{K = LY^{-1} : \text{(B.10) is feasible, } Y \succ 0\}. \end{aligned} \tag{B.11}$$

Noticing that the set of $(Y, L)$ satisfying (B.10) and $Y \succ 0$ is convex and hence path-connected. In addition, the map $K = LY^{-1}$ is continuous for positive definite $Y$. We can conclude that $\mathcal{K}_\gamma = \{K \in \mathcal{K} : J(K) \leq \gamma\}$ is path-connected. Such a proof is actually quite similar to the proof presented in [30]. The main difference is that the assumptions in this paper allow us to directly use the non-strict version of the bounded real lemma.

## B.3 Proof for Lemma 3

The proof of this lemma depends on the above convex parameterization $(Y, L)$. Since $\text{LMI}(Y, L, \gamma)$ is linear in $(Y, L, \gamma)$, the condition (B.10) is convex. Suppose we have $K = LY^{-1}$ where $(Y, L, J(K))$ is a feasible point for the convex regime $\text{LMI}(Y, L, J(K)) \preceq 0$ and $Y \succ 0$. In addition, by the non-strict version of the bounded real lemma, we know there exists a pair $(Y^*, L^*)$ such that $K^* = L^*(Y^*)^{-1}$, $\text{LMI}(Y^*, L^*, J(K^*)) \preceq 0$ and $Y^* \succ 0$. By convexity of the LMI condition $\text{LMI}(Y, L, \gamma) \preceq 0$ and $Y \succ 0$, we know that the line segment between $(Y, L, J(K))$ and $(Y^*, L^*, J(K^*))$ is also in this convex set. Therefore, for any $0 \leq t \leq 1$, we know the point $(Y + t\Delta Y, L + t\Delta L, J(K) + t(J(K^*) - J(K)))$ also satisfies

$$\text{LMI}(Y + t\Delta Y, L + t\Delta L, J(K) + t(J(K^*) - J(K))) \preceq 0, \ Y + t\Delta Y \succ 0$$

where $\Delta L = L^* - L$, and $\Delta Y = Y^* - Y$. Since $Y \succ 0$ and $Y^* \succ 0$, we automatically have $Y + t\Delta Y \succ 0$. Based on (B.11), we can construct a policy $(L + t\Delta L)(Y + t\Delta Y)^{-1}$ and the resultant closed-loop $\mathcal{H}_\infty$ norm must be smaller than or equal to $J(K) + t(J(K^*) - J(K))$. Formally, we have

$$J((L + t\Delta L)(Y + t\Delta Y)^{-1}) \leq J(K) + t(J(K^*) - J(K)).$$

Based on the fact $J(K^*) < J(K)$, we can use the above inequality to construct a direction $d$ such that $J'(K, d) < 0$. Specifically, let's choose $d = \Delta L Y^{-1} - LY^{-1}\Delta Y Y^{-1}$. Then we have

$$
\begin{aligned}
J'(K, d) &= \lim_{t \searrow 0} \frac{J(K + t(\Delta L Y^{-1} - LY^{-1}\Delta Y Y^{-1})) - J(K)}{t} \\
&= \lim_{t \searrow 0} \frac{J(LY^{-1} + t(\Delta L Y^{-1} - LY^{-1}\Delta Y Y^{-1})) - J(K)}{t} \\
&=_{(a)} \lim_{t \searrow 0} \frac{J((L + t\Delta L)(Y + t\Delta Y)^{-1}) - J(K)}{t} \\
&\quad + \lim_{t \searrow 0} \frac{J((L + t\Delta L)(Y + t\Delta Y)^{-1} + O(t^2)) - J((L + t\Delta L)(Y + t\Delta Y)^{-1})}{t} \\
&\leq_{(b)} \lim_{t \searrow 0} \left( \frac{J((L + t\Delta L)(Y + t\Delta Y)^{-1}) - J(K)}{t} + O(t) \right) \\
&\leq \lim_{t \searrow 0} \left( \frac{J(K) + t(J(K^*) - J(K)) - J(K)}{t} + O(t) \right) \\
&= J(K^*) - J(K) < 0,
\end{aligned}
$$

where the step $(a)$ relies on the fact that $(Y + t\Delta Y)^{-1} = Y^{-1} - tY^{-1}\Delta Y Y^{-1} + O(t^2)$ and the step $(b)$ uses the fact that $J(\cdot)$ is locally Lipschitz (Proposition 2). Finally, the last inequality holds since we know $J(K) > J(K^*)$. Notice $d \neq 0$. Otherwise the above argument still works and we have $J'(K, 0) < 0$. This is impossible since we know $J'(K, 0) = 0$. This leads to a contradiction, and we must have $d \neq 0$. This completes the proof for this lemma.

## B.4 Proof for Theorem 2

We first show that $K^n \in \mathcal{K}_{J(K^0)}$ for all $n$ by induction. By choice, we have $\delta^n = \frac{c\Delta_0}{n+1} \leq c\Delta_0$ for all $n$ with some $c \in (0, 1)$. For $n = 1$, we have $K^1 = K^0 - c\Delta_0 F^0 / \|F^0\|_2$. Since the norm of $F^0 / \|F^0\|_2$ is equal to one and $\delta^0 = c\Delta_0 > 0$, we have $K^1 \in \mathbb{B}_{\delta^0}(K^0)$, where $\mathbb{B}_{\delta^0}(K^0)$ is the $\delta^0$-ball centered at $K^0$. Based on the definition of $\Delta_0$, we know $\mathbb{B}_{\delta^0}(K^0) \subseteq \mathcal{K}$, and hence $K^1 \in \mathcal{K}$. In addition, (13) implies $J(K^1) \leq J(K^0) - \delta^0 \|F^0\|_2$. Hence we have $\mathcal{K}_{J(K^1)} \subseteq \mathcal{K}_{J(K^0)}$, which implies $K^1 \in \mathcal{K}_{J(K^0)}$. Similarly, we can repeat this argument to show $K^n \in \mathcal{K}_{J(K^0)}$ for all $n$.

Next, we can apply (11) to every step and then sum the inequalities over all $n$. Then the following inequality holds for all $N$:

$$\sum_{n=0}^{N} \delta^n \|F^n\|_2 \leq J(K^0) - J(K^{N+1}) \leq J(K^0) - J^*, \tag{B.12}$$

where the second inequality holds since $J(K^{N+1}) \geq J^*$. Since we have $\sum_{n=0}^{\infty} \delta^n = c\Delta_0 \sum_{n=1}^{\infty} \frac{1}{n} = +\infty$, we know $\liminf_{n \to \infty} \|F^n\|_2 = 0$. Therefore, there exists one subsequence

$\{n_i\}$ such that $\|F^{n_i}\|_2 \to 0$. For this subsequence, the policy parameter sequence $\{K^{n_i}\}$ stays in the compact set $\mathcal{K}_{J(K^0)}$. Hence we know that the resultant policy sequence $\{K^{n_i}\}$ is also bounded and has a convergent subsequence $\{K^{n_{i(l)}}\}$, which converges to some limit point $K^\infty \in \mathcal{K}_{J(K^0)}$. In addition, we have $\|F^{n_{i(l)}}\|_2 \to 0$ as well.

When $\delta > 0$ is sufficiently small, we know $\mathbb{B}_\delta(K^\infty) \subset \mathcal{K}$. Then there exists $N_\delta$ such that $\mathbb{B}_{\delta^{n_{i(l)}}}(K^{n_{i(l)}}) \subset \mathbb{B}_\delta(K^\infty)$ for all $n_{i(l)} \geq N_\delta$. Hence we have $F^{n_{i(l)}} \in \partial_{\delta^{n_{i(l)}}} J(K^{n_{i(l)}}) \subset \partial_\delta J(K^\infty)$ for all $n_{i(l)} \geq N_\delta$. Noticing that $F^{n_{i(l)}} \to 0$ and $\partial.J(\cdot)$ is closed, we have $0 \in \partial_\delta J(K^\infty)$ for any $\delta > 0$. It is also well-know that one has $\bigcap_{n_{i(l)}} \partial_{\delta^{n_{i(l)}}} J(K^\infty) = \partial_C J(K^\infty)$ (see Remark 2 for extra explanations). Hence we have $0 \in \partial_C(K^\infty)$, and $K^\infty$ has to be a Clarke stationary point. Based on Theorem 1, we have $J(K^\infty) = J^*$, and hence the function-value subsequence $\{J(K^{n_{i(l)}})\}$ converges to $J^*$. Notice that $\{J(K^n)\}$ is monotonically decreasing for the entire sequence $\{n\}$. Hence the sequence $\{J(K^n)\}$ has a limit, and this limit has to be $J(K^\infty) = J^*$. This completes the proof.

**Remark 2.** *For completeness, we also explain the well-known fact regarding $\bigcap_n \partial_{\delta^n} J(K) = \partial_C J(K)$ as $\delta^n \to 0$. Here $\{\delta^n\}$ is allowed to be any monotonically-decreasing sequence satisfying $\partial_{\delta^0} J(K) \subset \mathcal{K}$. By definition, we have $\partial_C J(K) \subset \partial_{\delta^n} J(K)$ for all $n$. Since $\{\delta^n\}$ is monotonically decreasing, we also have $\partial_{\delta^{n+1}} J(K) \subset \partial_{\delta^n} J(K)$ for all $n$. Therefore, we have*

$$\partial_C J(K) \subseteq \lim_{\delta^n \searrow 0} \partial_{\delta^n} J(K) = \bigcap_n \partial_{\delta^n} J(K).$$

*To show $\bigcap_n \partial_{\delta^n} J(K) \subset \partial_C J(K)$, we can use the following contradiction argument, which is standard (e.g. see [27, Remark 3.7]). Let us assume that there exists $V \in \bigcap_n \partial_{\delta^n} J(K) \setminus \partial_C J(K)$. Denote $S_n := \{\cup_{K' \in \mathbb{B}_{\delta^n}(K)} \partial_C J(K')\}$. Obviously, $S_n$ depends on $K$ and $\delta^n$. In [25], $S_n$ has been shown to be compact and nested. By [25, Lemma 2.6], we have*

$$V \in \bigcap_n \partial_{\delta^n} J(K) = \text{conv} \bigcap_n S_n.$$

*Therefore, we can express $V$ as $V = \sum_j t_j V_j$ with $V_j \in \bigcap_n S_n$, $t_j \geq 0$, and $\sum_j t_j = 1$. Notice $V_j \in S_n$ for all $n$. Based on the definition of Clarke subdifferential, we know that for each $n$, there exists a sequence of differentiable points $\{K_j^{n,r}\}$ such that $K_j^{n,r} \to K_j^n$, $\nabla J(K_j^{n,r}) \to V_j$ as $r \to \infty$, and $\|K_j^n - K\| \leq \delta^n$. Then there exists a large enough $r(n,j)$ such that $\|V_j - \nabla J(K_j^{n,r(n,j)})\| \leq \delta^n$, and $\|K_j^{n,r(n,j)} - K\| \leq 2\delta^n$. Since $\delta^n \to 0$ as $n \to \infty$, we have found a sequence $\{K_j^{n,r(n,j)}\}_{n=1}^\infty$, such that $K_j^{n,r(n,j)} \to K$ and $\nabla J(K_j^{n,r(n,j)}) \to V_j$ as $n \to \infty$. Therefore, we have $V_j \in \partial_C J(K)$. By convexity of $\partial_C J(K)$ [25, Proposition 2.3], we know $V = \sum_j t_j V_j \in \partial_C J(K)$. This contradicts the assumption that $V$ is not in $\partial_C J(K)$. Therefore, we must have $\bigcap_n \partial_{\delta^n} J(K) \subset \partial_C J(K)$. Consequently, we know $\bigcap_n \partial_{\delta^n} J(K) = \partial_C J(K)$.*

## C    Discussions on Implementable Variants and Related Convergence Results

In this section, we give more detailed discussions on implementable variants of Goldstein's subgradient method and related convergence guarantees. Formal treatments to the informal theorems presented in Section 4.2 are also presented.

### C.1    Gradient sampling

The gradient sampling (GS) method can be viewed as an approximated version of Goldstein's subgradient method. The idea of GS has been explained in Section 4.2. In the unconstrained optimization setting, it has been shown that every cluster point of GS can be guaranteed to be Clarke stationary (with probability one) [9, 10, 37]. For our problem, we need to ensure that the iterates do not travel outside the feasible set $\mathcal{K}$, and hence we use the trust-region version of GS, which was originally developed in [37, Section 4.2]. For clarity, the trust-region version of GS in [37, Section 4.2] is restated as Algorithm 1. For our purpose, we need to ensure $\delta^0 < \frac{\Delta_0}{2}$. See [37, Section 4.2] for more discussions on the convergence theory for Algorithm 1. As explained in [37, Section 4.2], Theorem 3.3 in [37] also holds for the trust-region version of GS, and hence we know that every cluster point of Algorithm 1 is Clarke stationary (with probability one). From

---

**Algorithm 1:** Gradient Sampling (GS)

---

**Require**: $K^0 \in \mathcal{K}$ at which $J$ is continuously differentiable, initial sampling radius $\delta^0 \in (0, \frac{\Delta_0}{2})$,
  initial stationarity target $\epsilon^0 \in [0, \infty)$, reduction factors $\mu_\delta, \mu_\epsilon \in (0, 1]$, sample size
  $m \geq n_x n_u + 1$, line search parameters $(\beta, \theta) \in (0, 1) \times (0, 1)$

**for** $n = 0, 1, 2, \cdots$ **do**
    Independently sample $\{K_{n,1}, \cdots K_{n,m}\}$ uniformly from $\mathbb{B}_{\delta^n}(K^n)$
    Compute $F^n$ as the solution of the following convex quadratic program:

$$\min \ \frac{1}{2}\|F\|_2^2$$

$$\text{subject to } F \in \mathcal{F}^n = \text{conv}\{\nabla J(K^n), \nabla J(K_{n,1}), \cdots, \nabla J(K_{n,m})\}$$

    **if** $\|F^n\|_2 \leq \epsilon^n$, set $\delta^{n+1} \leftarrow \mu_\delta \delta^n$, $\epsilon^{n+1} \leftarrow \mu_\epsilon \epsilon^n$, $t^n \leftarrow 0$, $K^{n+1} \leftarrow K^n$, and move to the
next round
    **else** set $\delta^{n+1} \leftarrow \delta^n$, $\epsilon^{n+1} \leftarrow \epsilon^n$, define:

$$\hat{F}^n = \delta^n F^n / \|F^n\|_2, \tag{C.1}$$

and compute $t^n$ such that

$$t^n \leftarrow \max\{t \in \{1, \theta, \theta^2, \cdots\} : J(K^n - t\hat{F}^n) < J(K^n) - \beta t \delta^n \|F^n\|_2\} \tag{C.2}$$

    **if** $J$ is continuously differentiable at $K^n - t^n \hat{F}^n$
        **then** set $K^{n+1} \leftarrow K^n - t^n \hat{F}^n$
        **else** set $K^{n+1}$ randomly as any point where $J$ is continuously differentiable such that

$$J(K^{n+1}) < J(K^n) - \beta t^n \delta^n \|F^n\|_2 \tag{C.3}$$

$$\|K^n - t^n \hat{F}^n - K^{n+1}\|_2 \leq \min\{t^n, \delta^n\} \|\hat{F}^n\|_2 \tag{C.4}$$

**end for**

---

the discussion in [37, Section 4.2], we also know that the trust-region version of GS can guarantee $\|K^{n+1} - K^n\| \leq 2t^n \delta^n \leq 2\delta^0 < \Delta_0$ for any $K^n \in \mathcal{K}$. Recall that $\Delta_0$ is the distance between $\mathcal{K}_{J(K^0)}$ and $\mathcal{K}^c$. Therefore, by induction, we can show that the iterations generated by Algorithm 1 will be guaranteed to stay in the sublevel set $\mathcal{K}_{J(K^0)}$ with probability one. Then it is straightforward to combine the above facts and Theorem 1 to show the global convergence of Algorithm 1 for the $\mathcal{H}_\infty$ state-feedback synthesis problem. We formalize Theorem 4 as the following global convergence result.

**Theorem C.1.** *Consider the policy optimization problem* (6) *with the $\mathcal{H}_\infty$ cost function defined in* (5). *Suppose Assumption 1 holds, and $K^0 \in \mathcal{K}$. Let $\{K^n\}$ be a sequence generated by Algorithm 1 with $\mu_\delta, \mu_\epsilon < 1$. With probability one, we have $K^n \in \mathcal{K}$ for all $n$, and the algorithm does not stop such that $\delta^n \downarrow 0$ and $\epsilon^n \downarrow 0$. In addition, the function-value sequence $\{J(K^n)\}$ is monotonically decreasing almost surely, and we have $J(K^n) \to J^*$ as $n \to \infty$ with probability one.*

*Proof.* As commented previously, we can use an induction argument to show that the iterates $\{K^n\}$ generated by Algorithm 1 stay inside the feasible set $\mathcal{K}$ almost surely for all $n$. Now we present some details for this argument. For $n = 0$, we know $\mathbb{B}_{\delta^0}(K^0) \subset \mathcal{K}$, and hence $\{K_{0,1}, \cdots, K_{0,m}\} \subset \mathcal{K}$. With probability one, $J$ is differentiable on $K_{0,j}$ for all $j \in \{1, 2, \cdots, m\}$. Therefore, $F^0$ is well defined with probability one. There are two possibilities. If $\|F^0\|_2 \leq \epsilon^0$, then we set $K^1 = K^0$ and shrink $(\epsilon^0, \delta^0)$. Obviously, we still have $K^1 \in \mathcal{K}_{J(K^0)}$. If $\|F^0\|_2 > \epsilon^0$, the algorithm is guaranteed to find a good descent condition satisfying (C.2). Notice that we have $K^0 - t\hat{F}^0 \in \mathcal{K}$ for any $t \in \{1, \theta, \theta^2, \cdots\}$. If $J$ is continuously differentiable at $K^0 - t^0 \hat{F}^0$, then we have $K^1 = K^0 - t^0 \hat{F}^0 \in \mathbb{B}_{\delta^0}(K^0) \subseteq \mathcal{K}$ as $\|\hat{F}^0\|_2 = \delta^0$ and $t^0 \leq 1$. If $J$ is not continuously differentiable at $K^0 - t^0 \hat{F}^0$, then $K^1$ is randomly generated in a way that (C.3) and (C.4) hold. From (C.4), we can combine $\delta^0 \in (0, \Delta_0/2)$ and $t^0 \leq 1$ to show $\|K^1 - K^0\| \leq 2t^0 \delta^0 < \Delta_0$. Then we have $K^1 \in \mathcal{K}$ as well. Since the line search guarantees $J(K^1) \leq J(K^0)$, we further have $K^1 \in \mathcal{K}_{J(K^0)}$. To summarize, no matter whether $\|F^0\|_2$ is larger than $\epsilon^0$ or not, we always have $K^1 \in \mathcal{K}_{J(K^0)} \subset \mathcal{K}$.

Then we can repeat this argument to show $K^n \in \mathcal{K}_{J(K^0)} \subset \mathcal{K}$ for all $n$ (with probability one). From [37, Section 4.2] (and Theorem 1), every cluster point of $\{K^n\}$ has to be a Clark stationary point (and hence a global optimal point) of $J$ in the almost sure sense[7]. From Lemma 2, $\mathcal{K}_{J(K^0)}$ is compact. Then with probability one, $\{K^n\}$ stays in the compact set $\mathcal{K}_{J(K^0)}$ for all $n$, and there must exist a subsequence of $\{K^n\}$ which converge to one of its cluster points. Therefore, this subsequence must converge to the global optimal point of $J$, and the function-value sequence associated with this policy subsequence has to converge to $J^*$ almost surely. Notice that $\{J(K^n)\}$ is monotonically decreasing for the entire sequence $\{n\}$ almost surely. Therefore, we have $J(K^n) \to J^*$ almost surely. $\qquad\square$

The implementation of (C.3) and (C.4) can be done via sampling as discussed in [37, Section 2]. In addition, $\Delta_0$ is typically unknown in practice, and one just needs to tune the parameter $\delta^0$ until it is sufficiently small. If the cost function is not continuously differentiable on the initial stabilizing policy, one can just randomly sample around that stabilizing policy with a sufficiently small sampling radius to obtain another stabilizing policy meeting the initialization requirement in Algorithm 1.

### C.2 Non-derivative sampling

The non-derivative sampling (NS) method was originally developed in [38], and can be viewed as the derivative-free version of GS. For the NS method, the gradient oracle in the GS algorithm is replaced by the zeroth-order oracle which is only required to return the function value $J(K)$ for given $K$, and similar convergence guarantees can still be obtained. To avoid the gradient evaluation, the Gupal estimation is used in the NS method to approximate the derivatives from function values. For completeness, we restate the NS method from [38] as follows. The mapping $\chi$ used in the algorithm description is given below by (C.5), and $\mathcal{Z}$ denotes the uniform distribution over the $n_u \times n_x$ unit cube, i.e. $[-1/2, 1/2]^{n_u \times n_x}$. Any $z \in \mathcal{Z}$ will be an $n_u \times n_x$ matrix.

---

**Algorithm 2:** Non-derivative Sampling (NS)

---

**Require**: initial stabilizing policy $K^0 \in \mathcal{K}$, initial sampling radius $\delta^0 \in (0, \Delta_0/2)$, initial stationarity target $\epsilon^0 \in [0, \infty)$, reduction factors $\mu_\delta, \mu_\epsilon \in (0, 1]$, sample size $m \geq n_x n_u + 1$, line search parameters $(\beta, \underline{t}, \kappa)$ in $(0, 1)$, and a sequence of positive mollifier parameters defined as $\alpha_n = \alpha_0/(n+1)$ with $\alpha_0 < \min\{\Delta_0/\sqrt{n_x n_u}, 1\}$.

**for** $n = 0, 1, 2, \cdots$ **do**

    Independently sample $\{K_{n,1}, \cdots K_{n,m}\}$ uniformly from $\mathbb{B}_{\delta^n}(K^n)$

    Independently sample $\{z_{n,1}, \cdots z_{n,m}\}$ uniformly from $\mathcal{Z}$

    Compute $F^n$ as the solution of

$$\min \ \frac{1}{2}\|F\|_2^2$$

    subject to $F \in \mathcal{F}^n = \text{conv}\{\chi(K_{n,1}, \alpha_n, z_{n,1}), \cdots, \chi(K_{n,m}, \alpha_n, z_{n,m})\}$

    **if** $\|F^n\| \leq \epsilon^n$, set $\epsilon^{n+1} \leftarrow \mu_\epsilon \epsilon^n$, $\delta^{n+1} \leftarrow \mu_\delta \delta^n$, $t^n \leftarrow 0$, $K^{n+1} \leftarrow K^n$, and move to the next round

    **else** set $\delta^{n+1} \leftarrow \delta^n$, $\epsilon^{n+1} \leftarrow \epsilon^n$, $\hat{F}^n \leftarrow F^n/\|F^n\|_2$, and $K^{n+1} \leftarrow K^n - t^n \hat{F}^n$, where $t^n$ is determined using the following line search strategy:

        (i) Choose an initial step size $t = t_{ini}^n = \delta^n \geq t_{\min}^n := \min\{\underline{t}, \kappa\delta^n/3\}$

        (ii) If $J(K^n - t\hat{F}^n) \leq J(K^n) - \beta t\|F^n\|$, return $t^n := t$

        (iii) If $\kappa t < t_{\min}^n$, return $t^n := 0$

        (iv) Set $t := \kappa t$, and go to (ii).

**end for**

---

For any $z \in \mathcal{Z}$, we denote the $(i, j)$-entry of $z$ as $z(i, j)$. For every $K \in \mathcal{K}$ and $z \in \mathcal{Z}$, we formally define $\chi(K, \alpha, z)$ to be a $n_u \times n_x$ matrix[8] whose $(i, j)$-th entry is calculated as

$$\chi_{ij}(K, \alpha, z) = \frac{1}{\alpha}\left(J(K + \alpha z + V_{ij}^+) - J(K + \alpha z + V_{ij}^-)\right), \tag{C.5}$$

---

[7] As pointed out in [9], the convergence theory for GS requires the cost function to be continuously differentiable over a set of full measure. Based on the specific form of the $\mathcal{H}_\infty$ cost (5) and the chain rule in [2], one can see that this is not an issue for the $\mathcal{H}_\infty$ state-feedback synthesis problem.

[8] Obvious, the dimensions of $\chi(K, \alpha, Z)$ and $K$ are the same.

where $V_{ij}^+ \in \mathbb{R}^{n_u \times n_x}$ is a matrix whose $(i,j)$-th entry is equal to $\frac{\alpha}{2} - \alpha z(i,j)$ and all other entries are 0, and $V_{ij}^- \in \mathbb{R}^{n_u \times n_x}$ is a matrix whose $(i,j)$-entry is equal to $-\frac{\alpha}{2} - \alpha z(i,j)$ and all other entries are 0. For our constrained optimization setup, the above definition assumes $K + \alpha z + V_{ij}^\pm \in \mathcal{K}$ for all $(i,j)$, and hence implicitly requires that $\alpha$ is small enough such that $K + [-\alpha/2, \alpha/2]^{n_u \times n_x} \subset \mathcal{K}$. Notice that $\chi(K, \alpha, z)$ is exactly the Gupal estimate for the gradient of some smoothed version (or formally the Steklov average) of $J(K)$. See [38, Section 2] for more detailed discussions. Based on the definition of $\chi(K, \alpha, z)$, the NS method can be implemented as described in Algorithm 2.

For Algorithm 2, we choose $\delta^0 \in (0, \Delta_0/2)$ and $\alpha_0 < \min\{\Delta_0/\sqrt{n_x n_u}, 1\}$ to ensure $F^n$ is well defined and $K^n \in \mathcal{K}$ almost surely for all $n$. With the help of [38, Theorem 3.8] and Theorem 1, we can establish the following global convergence result.

**Theorem C.2.** *Consider the policy optimization problem* (6) *with the $\mathcal{H}_\infty$ cost function defined in* (5). *Suppose Assumption 1 holds, and $K^0 \in \mathcal{K}$. Let $\{K^n\}$ be a sequence generated by Algorithm 2 with $\mu_\delta, \mu_\epsilon < 1$. For every step, $F^n$ is well defined. With probability one, $K^n \in \mathcal{K}$ for all $n$, and the algorithm does not stop such that $\delta^n \downarrow 0$ and $\epsilon^n \downarrow 0$. In addition, the function-value sequence $\{J(K^n)\}$ is monotonically decreasing almost surely, and we have $J(K^n) \to J^*$ as $n \to \infty$ with probability one.*

*Proof.* We can use an induction argument to show that $F^n$ is well defined, and $K^n \in \mathcal{K}$ almost surely. For $n = 0$, we know $\mathbb{B}_{\delta^0}(K^0) \subset \mathcal{K}$, and hence $\{K_{0,1}, \cdots, K_{0,m}\} \subset \mathcal{K}$. To ensure $F^0$ being well defined, we need $\chi(K_{0,l}, \alpha_0, z_{0,l})$ to be well defined for all $l \in \{1, 2, \cdots, m\}$. As discussed above, this can be guaranteed via ensuring $K_{0,l} + [-\alpha_0/2, \alpha_0/2]^{n_u \times n_x} \subset \mathcal{K}$ for all $l$. Since we have chosen $\alpha_0 < \Delta_0/\sqrt{n_x n_u}$ and $\delta^0 < \Delta^0/2$, we must have $K_{0,l} + [-\alpha_0/2, \alpha_0/2]^{n_u \times n_x} \subset \mathbb{B}_{\Delta_0/2}(K_{0,l}) \subset \mathbb{B}_{\delta^0 + \Delta^0/2}(K^0) \subset \mathcal{K}$. Therefore, $F^0$ is well defined. If $\|F^0\|_2 \leq \epsilon^0$, then we set $K^1 = K^0$ and shrink $(\epsilon^0, \delta^0)$. Obviously, we have $K^1 \in \mathcal{K}_{J(K^0)}$. If $\|F^0\|_2 > \epsilon^0$, we need to compute the step size $t^0$ as described in Algorithm 2. There are two possibilities: if $t^0 = 0$, then we still have $K^1 = K^0$ and hence $K^1 \in \mathcal{K}_{J(K^0)}$. If $t^0 \neq 0$, then the algorithm has found a good descent condition such that $J(K^0 - t^0 \hat{F}^0) \leq J(K^0) - \beta t^0 \|F^0\|$ holds. In this case, we set $K^1 = K^0 - t^0 \hat{F}^0$. Notice that we must have $K^0 - t^0 \hat{F}^0 \in \mathcal{K}$ as $t^0 \leq \delta^0$ and $\|\hat{F}^0\|_2 = 1$. Since the descent condition guarantees $J(K^1) \leq J(K^0)$, we further have $K^1 \in \mathcal{K}_{J(K^0)}$. To summarize, no matter whether $\|F^0\|_2$ is larger than $\epsilon^0$ or not, we always have $K^1 \in \mathcal{K}_{J(K^0)} \subset \mathcal{K}$. Then we can repeat this argument to show $F^n$ is well defined, and $K^n \in \mathcal{K}_{J(K^0)} \subset \mathcal{K}$ almost surely for all $n$. From [38, Theorem 3.8] (and Theorem 1), every cluster point of $\{K^n\}$ has to be a Clark stationary point (and hence a global optimal point) of $J$ in the almost sure sense. From Lemma 2, $\mathcal{K}_{J(K^0)}$ is compact, and $\{K^n\}$ must admit a subsequence which converge to one of its cluster points. Therefore, with probability one, this subsequence must converge to the global optimal point of $J$, and the function-value sequence associated with this policy subsequence has to converge to $J^*$. Notice that $\{J(K^n)\}$ is bounded and monotonically decreasing for the entire sequence $\{n\}$ in an almost sure sense. Therefore, we have $J(K^n) \to J^*$ with probability one. $\qquad\square$

Notice that $\Delta_0$ is typically unknown when implementing Algorithm 2. Hence one just tunes the values of $\delta^0$ and $\alpha^0$ until they are sufficiently small.

### C.3 Model-free NS

When the model is unknown, one can estimate the value of $J(K)$ from sampled trajectories of the closed-loop system, and implement a model-free version of NS via using the resultant stochastic zeroth-order oracle. As a matter of fact, there are many different methods available for estimating the $\mathcal{H}_\infty$-norm from data [45, 46, 57, 54, 69, 50, 67, 66]. For the model-free NS method, the exact function-value oracle for the $\mathcal{H}_\infty$ cost (5) is replaced by a stochastic oracle which relies on noisy estimates of the cost value. Based on our experience, the multi-input multi-output (MIMO) power iteration method [49] works quite well as a stochastic zeroth-order oracle for the purpose of implementing the model-free NS method. The main idea of the MIMO power iteration method is that the $\mathcal{H}_\infty$ norm of an LTI MIMO system can be estimated from the largest singular value of its finite-time approximated representation (which can be thought as a matrix), and a specialized time reversal trick can be used to make the computation efficient. Given a black-box simulator for a stable system $G_K$, the MIMO power iteration method provides a reasonably good oracle for estimating

the $\mathcal{H}_\infty$ norm of $G_K$ from sampled trajectories. It is worth mentioning that the estimation quality depends on the length of the time window over which $G_K$ is approximated. We denote this window length as $N$. The larger $N$ is, the better the $\mathcal{H}_\infty$-norm estimation is. We refer the readers to [49] for implementation details of the MIMO power iteration method.

The sample complexity for model-free NS is still unknown. However, our numerical results show that such a model-free method tracks the convergence of its model-based counterpart given reasonable amount of data[9].

### C.4 Interpolated Normalized Gradient Descent (INGD) with finite-time complexity

Both GS and NS methods do not have finite-time complexity guarantees for finding $(\delta, \epsilon)$-stationary points. The INGD method was originally proposed in [71], and provide an alternative implementable approximation for Goldstein's subgradient method. The main advantage of INGD is that it yields finite-time iteration complexity for finding $(\delta, \epsilon)$-stationary points. The original version of INGD developed in [71] requires a special generalized gradient oracle (see [71, Assumption 1] for details). To relax this requirement, [14] proposed a variant of INGD which only requires standard gradient oracle for any differentiable points. Specially, at an iterate $K^n$, the INGD method in [14] relies on Algorithm 3 to compute the update direction $F^n$. Again, we slightly abuse our notation by using $L$ to denote the Lipschitz constant appearing in Algorithm 3 (previously, we used $L$ to denote a particular matrix in the LMI formulation for $\mathcal{H}_\infty$ state-feedback synthesis). In the unconstrained setup, it has been shown that Algorithm 3 terminates and generates a good descent direction with high probability [14, Corollary 2.5]. Then one can combine Algorithm 3 and (13) to formulate the INGD method, which is formally given as Algorithm 4. In the unconstrained optimization setting, finite-time iteration complexity for finding $(\delta, \epsilon)$-stationary points have been obtained for Algorithm 4 [14, Theorem 2.6].

---

**Algorithm 3:** MinNorm

---

**Input**: $K$, $\delta > 0$, $\epsilon > 0$, and the Lipschitz constant $L$
Set $F = \nabla J(\Xi)$ where $\Xi$ is sampled uniformly from $\mathbb{B}_\delta(K)$
**while** $\|F\|_2 > \epsilon$ **and** $\frac{\delta}{4}\|F\|_2 \geq J(K) - J(K - \delta F/\|F\|_2)$ **do**

    Choose any $r$ satisfying $0 < r < \|F\|_2\sqrt{1 - (1 - \frac{\|F\|_2^2}{128L^2})^2}$
    Sample $\Upsilon$ uniformly from $\mathbb{B}_r(F)$
    Sample $\Xi$ uniformly from $[K, K - \delta\frac{\Upsilon}{\|\Upsilon\|_2}]$
    $F \leftarrow \arg\min_{\Phi \in [F, \nabla J(\Xi)]} \|\Phi\|_2$
**end while**
Return $F$

---

---

**Algorithm 4:** Interpolated Normalized Gradient Descent (INGD)

---

**Initial**: $K^0$, $T$
**Input**: $\delta$, $\epsilon$, and the Lipschitz constant $L$
**for** $n = 0, 1, \cdots, T$ **do**
    $F^n = \text{MinNorm}(K^n, \delta, \epsilon, L)$
    $K^{n+1} = K^n - \delta F^n/\|F^n\|_2$
**end**
Return $K^T$

---

Now we discuss how to modify the finite-time analysis in [14] for our $\mathcal{H}_\infty$ control problem. It has been shown in [14, Theorem 2.6] that for unconstrained nonsmooth optimization of $L$-Lipschitz functions, the above INGD algorithm can be guaranteed to find one $(\delta, \epsilon)$-stationary point with the high-probability iteration complexity $\mathcal{O}\left(\frac{\Delta L^2}{\epsilon^3 \delta} \log(\frac{\Delta}{p\delta\epsilon})\right)$, where $\Delta$ is the initial function value gap, and $p$ is the failure probability (i.e. the optimization succeeds with the probability $(1 - p)$). We can choose $\delta < \Delta_0$ to ensure that the iterates from Algorithm 4 are well defined and stay in the

---

[9]Specifically, $N$ should be chosen properly and cannot be too small.

Table 1: Algorithm parameters for GS and NS

| Parameter | $(n_x, n_u, m)$ | $\delta^0$ | $\epsilon^0$ | $\mu_\delta$ | $\mu_\epsilon$ | $\delta_{opt}$ | $\epsilon_{opt}$ | $\beta$ | $\theta$ | $\alpha_n$ |
|---|---|---|---|---|---|---|---|---|---|---|
| GS | $(3, 1, 4)$ | 0.01 | 100 | 0.5 | 0.5 | 0 | 0 | 0.5 | 0.9 | N/A |
| NS | $(3, 1, 4)$ | 0.01 | 100 | 0.5 | 0.5 | 0 | 0 | 0.5 | 0.9 | $0.1/(n+1)$ |

feasible set $\mathcal{K}$ almost surely, and this immediately leads to the desired conclusion that the above INGD algorithm can find the $(\delta, \epsilon)$-stationary point for the $\mathcal{H}_\infty$ state-feedback synthesis problem with the same high-probability iteration complexity $\mathcal{O}\left(\frac{\Delta L^2}{\epsilon^3 \delta} \log(\frac{\Delta}{p\delta\epsilon})\right)$. Now we formalize Theorem 5 as follows.

**Theorem C.3.** *Consider the $\mathcal{H}_\infty$ state-feedback control problem* (6) *with the objective function $J(K)$ defined in* (5). *Suppose Assumption 1 holds, and the initial policy is stabilizing, i.e. $K^0 \in \mathcal{K}$. Denote $\Delta_0 := \text{dist}(\mathcal{K}_{J(K^0)}, \mathcal{K}^c)$. Let $\Delta = J(K^0) - J^*$. Denote the Lipschitz constant of $J(K)$ over the sublevel set $\mathcal{K}_{J(K^0)}$ as $L_0$. For any $\delta < \Delta_0$, the iterations generated by the INGD algorithm will stay in $\mathcal{K}$ almost surely. In addition, the INGD method finds a $(\delta, \epsilon)$-stationary point with a high-probability iteration complexity $\mathcal{O}\left(\frac{\Delta L_0^2}{\epsilon^3 \delta} \log(\frac{\Delta}{p\delta\epsilon})\right)$, where $p$ is the failure probability.*

*Proof.* We first use an induction argument to show that the iterates of INGD are well defined and stay inside the feasible set $\mathcal{K}$ (almost surely) when $\delta < \Delta_0$. For $n = 0$, notice that the variable $\Xi$ (used in Algorithm 3) has to be in $\mathcal{K}$ almost surely and well defined. The reason is that we sample $\Xi$ uniformly from $[K, K - \delta^0 \frac{\Upsilon}{\|\Upsilon\|_2}]$ and $\delta^0 < \Delta_0$. Next, the update direction $F^0$ satisfies either $\|F^0\|_2 \le \epsilon$ or $J(K^0 - \delta^0 F^0/\|F^0\|_2) \le J(K^0) - \frac{\delta^0}{4}\|F^0\|_2$. If $\|F^0\| \le \epsilon$, the INGD algorithm will terminate as we have found a $(\delta, \epsilon)$-stationary point. If $\|F^0\| > \epsilon$, we have $K^1 = K^0 - \delta^0 F^0/\|F^0\|_2$. We must have $K^1 \in \mathcal{K}$ as well since $\delta^0 < \Delta_0$. In addition, since $J(K^1) \le J(K^0)$, we have $K^1 \in \mathcal{K}_{J(K^0)} \subset \mathcal{K}$. Then we can repeat this argument to show $K^n \in \mathcal{K}_{J(K^0)}$ for all $n$ almost surely, and the high probability complexity bound becomes just a direct consequence of [14, Theorem 2.6]. $\square$

# D   Numerical Experiments

In this section, we provide simulation results to support our theory. The simulations are executed on a desktop computer with a 3.3 GHz Intel Xeon W-1350 processor, and the implementation is done using MATLAB R2021b. We tested GS, NS, model-free NS, and INGD on several examples, and the details are given below.

### D.1   More details on the simulation results in Figure 1

**Implementation of INGD, GS, NS, and model-free NS methods:** For the problem matrices given in (16), we have $J^* = 7.3475$. Now we discuss how we obtain the left plot in Figure 1. In order to obtain the Clark stationary point by using INGD method, we need to run the INGD algorithm with $\delta$ decreasing to 0. To this end, we start with $\epsilon = 1 \times 10^{-5}$ and $\delta = 0.01$. Whenever the INGD method successfully finds a $(\delta, \epsilon)$-stationary point, we reduce $\delta$ by a constant factor 0.7. As $\delta$ decreases to 0, one should expect that the INGD algorithm approaches a Clark stationary point. In addition, we also implement the GS and NS methods, where the algorithm parameters are set as in Table 1. Finally, to implement the model-free NS method for the given problem matrices (16), we choose the sample size in the $\mathcal{H}_\infty$-norm estimation oracle as $N = 100$ (this just means that we approximate the LTI system over a 100-step time window).

**Implementation of randomly generated cases**: The middle plot of Figure 1 demonstrates the performance of the NS algorithm on some randomly generated cases. In particular, we set $A \in \mathbb{R}^{3\times3}$ to be $I + \xi$, where each element of $\xi \in \mathbb{R}^{3\times3}$ is sampled uniformly from $[0, 1]$. We set $B \in \mathbb{R}^{3\times1}$ with each element uniformly sampled from $[0, 1]$. We set $Q = I + \zeta I \in \mathbb{R}^{3\times3}$ with $\zeta$ uniformly sampled from $[0, 0.1]$. We set $R \in \mathbb{R}$ uniformly sampled from $[1, 1.5]$. For each experiment, the initial condition $K^0 \in \mathbb{R}^{1\times3}$ is also randomly sampled such that $\rho(A - BK^0) < 1$.

**Implementation of Model-free NS**: The right plot of Figure 1 provides the simulation results of the model-free NS method with different choices of sample size $N$ used in the $\mathcal{H}_\infty$-norm estimation

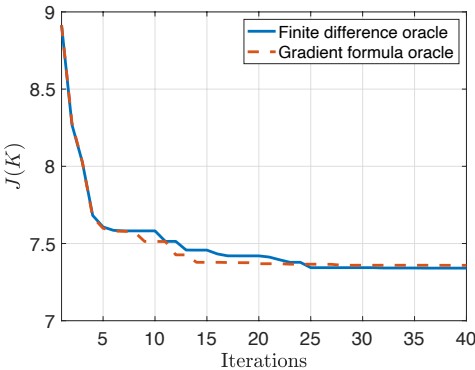 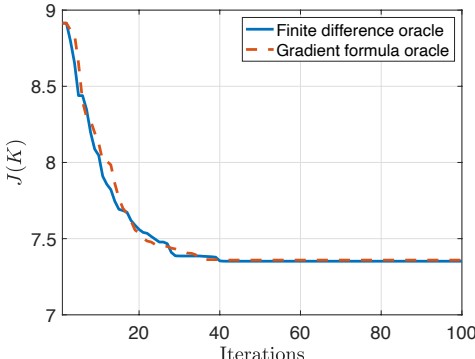

Figure 2: Simulation result for finite-difference oracle and gradient formula oracle. Left: INGD method. Right: GS method.

oracle. The hyperparameters of the model-free NS method are set to be the same as the ones used in NS. The only new issue is that we need to specify a sample size in the MIMO power iteration method for evaluating $J(K)$ from sampled trajectories. Specifically, we choose $N = [100\ 50\ 20\ 10]$. As we can see from the right plot in Figure 1, as we decrease the sample size, the $\mathcal{H}_\infty$-norm oracle become more noisy, and the algorithm performance has been degraded with more oscillations.

**Gradient oracle for differentiable points.** To implement INGD and GS, we need an oracle which is capable of computing the gradient of $J(K)$ for any differentiable points. We provide two options for implementing such gradient oracle. For the first option, we can just use a finite-difference scheme to estimate the gradient at any differentiable points. Specifically, for any differentiable point $K$, we can estimate $\nabla J(K)$ as a matrix whose $(i,j)$-th entry is computed as $\frac{J(K+hV_{ij})-J(K-hV_{ij})}{2h}$, where $V_{ij} \in \mathbb{R}^{n_u \times n_x}$ is a matrix whose $(i,j)$-th entry is 1 and all other entries are 0, and $h$ is a small positive parameter (e.g. $h = 0.0001$). For the second option, we can compute the gradient at differentiable points using some explicit analytic formula based on singular value decomposition. Specifically, we can use a special case of the subgradient analytical formula provided in [2]. To this end, let us rewrite $J(K)$ as follows:

$$J(K) = \sup_{\omega \in [0, 2\pi]} \sigma_{\max}(H(K, \omega)), \tag{D.1}$$

where $H(K, \omega) = (Q + K^\mathsf{T} RK)^{\frac{1}{2}}(e^{j\omega}I - A + BK)^{-1}$ and $\sigma_{\max}$ denotes the maximum singular value. Notice that $J(K)$ is differentiable at $K$ if the $\mathcal{H}_\infty$ norm of $H(K, \omega)$ is achieved at one frequency $\omega_0$ and the largest singular value of $H(K, \omega_0)$ has multiplicity one [63]. Then we can perform the singular value decomposition of $H(K, \omega_0)$ and take $u_1$ (which is the unit left-singular vector) and $v_1$ (which is the unit right-singular vector) corresponding to the largest singular value. Denote $H_1(K) = (Q + K^\mathsf{T} RK)^{\frac{1}{2}}$, $H_2(K) = (e^{j\omega_0}I - A + BK)^{-1}$, and define $\Gamma = \int_0^\infty e^{-\tau H_1(K)}H_2(K)v_1u_1^* e^{-\tau H_1(K)}d\tau$. Then the gradient of $J(K)$ at differentiable point $K$ can be calculated as $\nabla J(K) = \mathrm{Re}(RK(\Gamma + \Gamma^\mathsf{T}) - (H_2(K)v_1u_1^* H(K, \omega_0)B)^\mathsf{T})$. The derivation of the above formula uses the chain rule of the total derivative, and can be viewed as a special case of the general subgradient formula given in [2]. The two options lead to similar performances, as shown in Figure 2. We tested both options for implementing INGD and GS with the problem matrices given in (16). From Figure 2, we can see that both options work well and generate similar trajectories.

### D.2   Numerical results for INGD with constant choices of $(\delta, \epsilon)$

The simulation results for INGD with constant choices of $(\delta, \epsilon)$ are shown in Figure 3, where we set $\delta = 0.01$ and $\epsilon = 1 \times 10^{-8}$. From the left figure of Figure 3, it can be seen that it takes 10 steps for INGD to find a $(\delta, \epsilon)$-stationary point. At step 10, we have $\|F\|_2 < \epsilon$. However, $J(K)$ does not converge to the optimal value as shown in the right plot of Figure 3. This result confirms that $(\delta, \epsilon)$-stationarity does not imply being $\delta$-close to an $\epsilon$-stationary point of $J$ as commented in [71, 14].

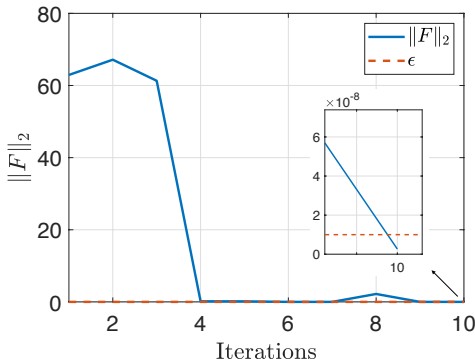 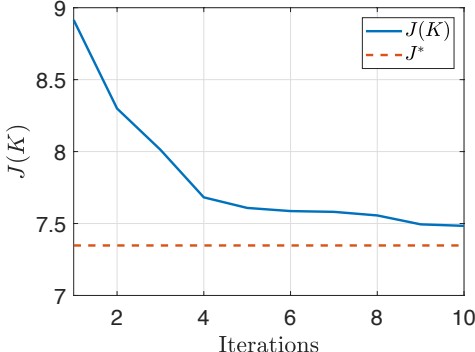

Figure 3: Simulation result for one-time INGD methods with $\delta = 0.01$ and $\epsilon = 1 \times 10^{-8}$. Left: the trajectory of the $\|F\|_2$. Right: the trajectory of function value $J(K)$.

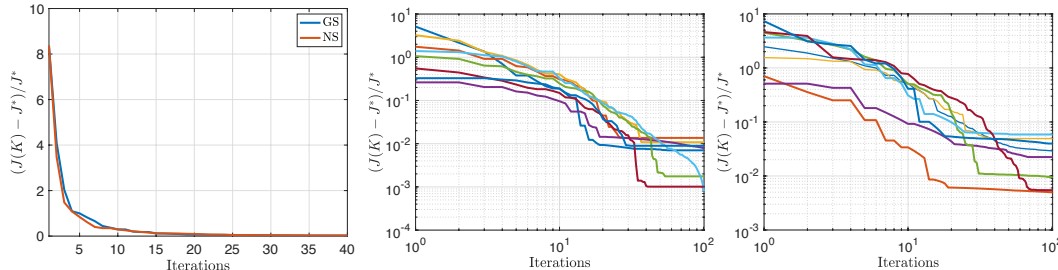

Figure 4: Simulation results for higher dimensional cases. Left: The trajectory of relative error of NS with problem matrices (D.2). Middle: The trajectory of relative error of 8 randomly generated cases for NS method. Right: The trajectory of relative error of 8 randomly generated cases for GS method.

### D.3 More examples

We further tested the NS and GS methods on some slightly larger examples. In particular, we first consider the following set of problem matrices:

$$A = \begin{bmatrix} 1.7865 & 0.3912 & 0.8758 & 0.5996 \\ 0.2756 & 1.3175 & 0.7692 & 0.4848 \\ 0.4764 & 0.9786 & 1.0618 & 0.7591 \\ 0.4489 & 0.7918 & 0.6014 & 1.7520 \end{bmatrix}, \ B = \begin{bmatrix} 0.1303 & 0.0312 \\ 0.1309 & 0.0528 \\ 0.7452 & 0.6727 \\ 0.2460 & 0.0743 \end{bmatrix}, \tag{D.2}$$

$$Q = 1.0613 I_4, \ R = 1.1315 I_2.$$

For the above example, we have $J^* = 43.26$ and we select the initialization $K^0$ to be:

$$K^0 = \begin{bmatrix} 2.4364 & 2.2337 & 2.4867 & 1.5551 \\ 12.1213 & -4.6823 & 2.1718 & -2.5906 \end{bmatrix},$$

which satisfies $\rho(A - BK^0) = 0.9567 < 1$. The results are reported in the left plot of Figure 4.

In addition, we also perform the NS and GS algorithm on some randomly generated cases. Similarly, we set $A \in \mathbb{R}^{4 \times 4}$ to be $I + \xi$, where each element of $\xi \in \mathbb{R}^{4 \times 4}$ is sampled uniformly from $[0, 1]$. We set $B \in \mathbb{R}^{4 \times 2}$ with each element uniformly sampled from $[0, 1]$. We set $Q = I + \zeta I \in \mathbb{R}^{4 \times 4}$ with $\zeta$ uniformly sampled from $[0, 0.1]$, and $R = I + \upsilon I \in \mathbb{R}^{2 \times 2}$ with $\upsilon$ uniformly sampled from $[0, 0.5]$. For each experiment, the initial condition $K^0 \in \mathbb{R}^{2 \times 4}$ is also chosen such that $\rho(A - BK^0) < 1$.

Figure 4 shows the simulation results on these examples. The left plot demonstrates the convergence of NS and GS methods with the problem matrices in (D.2). The middle and right plots demonstrate the performance of NS and GS on the randomly generated examples, respectively. It can be seen that both GS and NS work quite well on these examples.

# E    Further Discussions

## E.1    Uniqueness of the minimizing set

Our theory does not answer whether the global minimum for the $\mathcal{H}_\infty$ state-feedback synthesis problem is unique or not. As commented in Section 3, the path-connectedness of $\mathcal{K}_\gamma$ for every $\gamma$ can be used to show that there exists a unique global minimizing set in a certain sense [42, Sections 2&3]. However, we are not able to rule out the possibility that the uniuqe global minimizing set actually consists of multiple points. Whether the global minimum of the $\mathcal{H}_\infty$ state-feedback control problem is unique or not is an interesting open question.

## E.2    Possible generalizations for nonlinear systems

For nonlinear systems, it is possible to generalize Theorems 3 and 5. The proofs for the finite-time complexity of finding $(\delta, \epsilon)$-stationary points can be generalized to the constrained policy optimization setting, as long as the cost function $J$ is coercive over the nonconvex feasible set $\mathcal{K}$. In contrast, the convergence to global minimum may be too much to ask for general nonlinear robust control problems. If the cost function $J$ is not coercive, one may just add regularization to induce coerciveness such that one can still find some approximated $(\delta, \epsilon)$-stationary points provably. Such developments are worth more investigations in the future.