# OpenReview forum: "Global Convergence of Direct Policy Search for State-Feedback $\mathcal{H}_\infty$ Robust Control: A Revisit of Nonsmooth Synthesis with Goldstein Subdifferential"
_NeurIPS.cc/2022/Conference — NeurIPS 2022 Accept_

### Official Review · Reviewer_fj7B · 2022-06-21

**Rating:** 7
**Confidence:** 3
**Soundness:** 3 good
**Presentation:** 3 good
**Contribution:** 2 fair

**Summary:**

The authors study direct policy optimization for finding the minimal $\mathcal{H}_\infty$-norm stabilizing controller. They prove various analytic properties about the landscape of policy optimization such that a Goldstein subdifferential method converges in function value to the global minimum. While the Goldstein subdifferential method can be difficult to implement, they also discuss various sampling based strategies for finding the minimum norm subdifferential.

**Questions:**

- As a point of clarification, is there a unique minimum $K^*$ for problem (6), or are there possibly many global minima?

- Can the authors discuss the barriers to removing the $Q$ positive definite assumption and replacing it with a more standard $(A, Q^{1/2})$ is observable assumption?

- For the proof of Theorem 1, I am curious why the first case assuming $J’(K^{\dagger}, d) > 0$ is necessary. What is insufficient with the following argument: suppose $K^{\dagger}$ is a Clarke stationary point, but $K^{\dagger}$ is not a global minimum and hence $J(K^{\dagger}) > J^*$. In this case, Lemma 3 guarantees the existence of $d \neq 0$ such that $J’(K^{\dagger}, d) < 0$. However, by the subdifferential regularity of $J$ (Proposition 2), we know that $J’(K^{\dagger}, d) \geq 0$ for all $d$, hence a contradiction.

- Theorem 2: consider stating that $\delta^n = \frac{c \Delta_0}{n+1}$ for some $c \in (0, 1)$ instead of setting $c = 0.99$. The proof still goes through, but this would allow you to underestimate $\Delta_0$.

- Theorem 2: how do you compute (or estimate) $\Delta_0$?

- Theorem 2: Does $K^n \rightarrow K_\infty$ with $K_\infty \in \mathcal{K}$? The theorem is a statement about the function values, but is it possible for the $K^n$’s to not converge? This is related to my question about uniqueness of $K^*$.

- Theorem 2: Can one obtain a rate of convergence on $J(K^n) \rightarrow J^*$?


**Limitations:**

Some limitations are already discussed in weaknesses.

Also, the authors point out that their work only applies to the full state observation case, with partial observability needing future work. This is not an issue for me, as I see the partial observability case as follow up work.

**Strengths And Weaknesses:**

**Strengths:**
- The paper is written very clearly and concisely.
- The result is elegant.
- The proofs are clean and read quite nicely.

**Weaknesses:**

- The main result is asymptotic (Theorem 2 does not give any rates of convergence).

- It is acknowledged that computing the minimum norm subdifferential is challenging (see e.g. Line 317), but the sampling algorithms to address this are discussed mainly in the appendix. The authors present several implementation schemes, but it is not clear as a reader which one to prefer.

- An end to end statement combining Theorem 2 and any one of the sampling schemes to give a result on a practical algorithm is not clearly stated in the paper (although can possibly be inferred from the discussion in Appendix C).

- The algorithms listed in the appendix (Algorithm 1,2,3,4) all require function evaluations of the form $J(K)$, which involves computing the $\mathcal{H}_\infty$-norm of the system in feedback with a given $K$. Without access to the system dynamics $(A, B)$ (which is the whole point of studying policy search methods otherwise the $K^*$ can be computed via an SDP as noted in Lines 124-126), it is not clear how one computes $J(K)$, even with access to a simulator.

- There are no experiments showing the effectiveness of the proposed algorithms.

**Recommendation:**

In light of these weaknesses, I am not sure about the immediate impact/relevance of this paper beyond a small niche in the NeurIPS community. Nevertheless, it is a nice result from a mathematical point of view which in my opinion warrants an accept. The one caveat is that I am not that familiar with the nonsmooth optimization literature. Hence, while the results appear novel to me, if a reviewer who is more knowledgable in this area disagrees, then I would reconsider my recommendation based on said reviewer's arguments.

**EDIT:** After reviewing the author's responses and draft, I raised my score from 6 to 7.

---

> ### Author Response · Authors · 2022-08-02
> **Response to Reviewer fj7B: Part I**
>
> We really appreciate the reviewer for the constructive suggestions. We respond to your concerns as below.
>
> ***The main result is asymptotic (Theorem 2 does not give any rates of convergence.***
>
> As commented in the general response, the complexity for nonsmooth optimization of Lipschitz functions is quite subtle. The key subtlety is that  $(\delta,\epsilon)$-stationarity does not imply being $\delta$-close to an $\epsilon$-stationary point of $J$, and the finite-time complexity for finding $(\delta,\epsilon)$-stationary points does not imply too much about the convergence rate of the function values.
> We briefly discussed the complexity of finding $(\delta,\epsilon)$-stationary points for the INGD algorithm in the supplementary materail of our original submission. To clarify this issue, we add Theorem 3 and Theorem 5 to our revised paper which provides the finite-time complexity bounds for finding $(\delta,\epsilon)$-stationary points. While such results give a reasonable characterization of the finite time performance of the Goldstein subdifferential method and the INGD method on the $\mathcal{H}_\infty$ state-feedback control problem, they do not quantify how fast $J(K^n)$ converges to $J^*$. Recall that $(\delta,\epsilon)$-stationarity means  $\delta$-Goldstein subdifferential includes an element with norm being smaller than or equal to $\epsilon$, while $\epsilon$-stationarity means the Clarke subdifferential includes an element with norm being smaller than or equal to $\epsilon$. It is well-known that  $(\delta,\epsilon)$-stationarity does not imply being $\delta$-close to an $\epsilon$-stationary point of $J$. Consequently, the function value of a $(\delta,\epsilon)$-stationary point can be very far from $J^*$ even for very small $\delta$ and $\epsilon$. Theorem 5 in Ref [58] of our revised paper shows that there is no finite time algorithm that can find $\epsilon$-stationary points provably for all Lipschitz functions. It is still possible that one can develop some finite time  bounds for $(J(K^n)-J^*)$ via exploiting other advanced properties of the $\mathcal{H}_\infty$ cost function. This is an important future task.
>
> ***Computing the minimum norm in subdifferential is challenging, but the sampling algorithms are discussed mainly in the appendix. The authors present several implementation schemes, but it is not clear as a reader which one to prefer.***
>
> We revise our main paper to address this comment. In Section 4.2 of our revised paper, we discuss four algorithms: gradient sampling (GS), nonderivative sampling (NS), model-free NS, and Interpolated normalized gradient descent (INGD). GS, NS, and INGD are well-known variants of the Goldstein subdifferential method in the unconstrained setting. We adapt these algorithms to our constrained setting. Advantages and disadvantages are summarized below.
>
> 1. GS requires a first-order oracle to return $\nabla J(K)$ for any differentiable points in $\mathcal{K}$. From existing literature on GS, it is known that every cluster point of GS is Clarke stationary (with probability one). We can combine this result with our results (Lemma 4 and Theorem 1) to show that GS has an asymptotic global convergence guarantee. See Theorem 4 and a formal treatment in the supplementary material. The advantage of this method is that it was developed early and has already been included in the robust control toolbox HIFOO. So if one prefers to directly use a toolbox in a model-based setting, this method is easy to use.
>
> 2. NS is the derivative-free version of GS, and only requires a zeroth-order oracle to return $J(K)$ for any points in $\mathcal{K}$. From the existing literature on NS, it is known that every cluster point of NS is Clarke stationary (with probability one). We can combine this result with our results (Lemma 4 and Theorem 1) to show that GS has an asymptotic global guarantee. This is discussed in the revised version of our supplementary material. Compared to model-free NS, this method has an asymptotic guarantee.
>
> 3. Model-free NS: When the model is unknown, one has to estimate $J(K)$ using existing estimators for $\mathcal{H}_\infty$ norms. There exist many options. Based on our experience, the MIMO power iteration method (Ref [42] from our revised paper) works well for the purpose of implementing the NS method in the model-free setting. Currently, there is no sample complexity for model-free NS, but this will be our main choice of algorithm when we need to solve the problem in a model-free manner. Our numerical results in Figure 1 show that model-free NS closely tracks the convergence of its model-based counterpart given reasonable amount of samples.
>
> 4. INGD: This method only requires a first-order oracle to return $\nabla J(K)$ for any differentiable points in $\mathcal{K}$. The advantage of this method is that it yields a finite-time complexity bound for finding $(\delta, \epsilon)$-stationary points. See Theorem 5.

---

> > ### Author Response · Authors · 2022-08-02
> > **Response to Reviewer fj7B: Part II**
> >
> > ***An end to end statement combining Theorem 2 and any one of the sampling schemes to give a result on a practical algorithm is not clearly stated in the paper (although can possibly be inferred from the discussion in Appendix C).***
> >
> > Several end-to-end results (Theorems 4 and 5) have been added to the revised paper.
> >
> >
> > ***The algorithms listed in the appendix (Algorithm 1,2,3,4) all require function evaluations of the form $J(K)$, which involves computing the $\mathcal{H}_\infty$-norm of the system in feedback with a given $K$. Without access to the system dynamics $(A,B)$, it is not clear how one computes $J(K)$, even with access to a simulator.***
> >
> > Thanks for this useful comment. There are many estimation methods for estimating $\mathcal{H}_\infty$-norms from data generated by a simulator. We have added such a discussion to the last paragraph in Page 8 of our revised paper which discusses the model-free implementation of the NS method. Based on our own experience, the MIMO power iteration method works well as an $\mathcal{H}_\infty$-norm estimator for the purpose of implementing model-free NS. So we use this method in implementing the model-free NS method. Some results have been reported in Figure 1. The proposed method works reasonably well for these examples although we do  not have sample complexity bounds for model-free NS at this moment.
> >
> > ***There are no experiments showing the effectiveness of the proposed algorithms.***
> >
> > We have added numerical experiments to our revised paper to show how the implemental variants of the Goldstein subdifferential method work in simulations.
> >
> > ***As a point of clarification, is there a unique minimum  for problem (6), or are there possibly many global minima?***
> >
> > This is an open question. We can prove there is a unique global minimizing set (see Sections 2&3 in Ref[36] of our revised paper). However, right now our theory cannot answer whether the global minimum is unique or not.
> >
> > ***Can the authors discuss the barriers to removing the $Q$ positive definite assumption and replacing it with a more standard  observable assumption?***
> >
> > The positive definiteness of $Q$ is important for proving coercivity of $J(K)$. This is similar to the LQR case. The coercivity of the LQR cost also requires such an assumption on $Q$. This is the reason why such an assumption is made in the first few papers on global convergence of policy optimization on LQR. In the future, it is possible to develop new analysis which does not rely on the coercivity property such that the $Q$ positive definite assumption can be relaxed. That is an interesting future direction.
> >
> > ***Simplifying the proof of Theorem 1***
> >
> > Yes, thanks for the suggestion. The proof for Theorem 1 can be simplified exactly as suggested by the reviewer. We have made that change in our revised paper.
> >
> > ***choosing $c\in (0,1)$***
> >
> > The reviewer is correct. We have revised our paper accordingly by replacing $0.99$ with this constant $c$.
> >
> > ***Theorem 2: how do you compute (or estimate) $\Delta_0$?***
> >
> > When implementing the proposed algorithm, one does not need to compute $\Delta_0$. Theorem 2 just states that if one initialize the learning rate from some sufficiently small number, then the iterates can be guaranteed to stay in the feasible set and achieve global convergence. For actual implementation, one just tunes the initial stepsize in a heuristic way. If the algorithm does not work, just decrease the stepsize until the algorithm starts to generate global convergence behaviors.
> >
> > ***Does $K^n$ converge?***
> >
> > This is an open question. Right now we cannot rule out the possibility that different subsequences of $\{K^n\}$ actually converge to different global minimum in the global minimizing set. As mentioned previously, it is unclear whether the global minimum is unique or not in the first place. Thanks to the monotonicity of $J(K^n)$, we can prove that $J(K^n)$ converges to $J^*$.
> >
> >
> > ***Theorem 2: Can one obtain a rate on how fast $J^n$ converges?***
> >
> > Right now we do not have such a bound.
> > As commented before,  $(\delta,\epsilon)$-stationarity does not imply being $\delta$-close to an $\epsilon$-stationary point of $J$. Consequently, the function value of a $(\delta,\epsilon)$-stationary point can be very far from $J^*$ even for very small $\delta$ and $\epsilon$. Theorem 5 in Ref [58] of the revised paper shows that there is no finite time algorithm that can find $\epsilon$-stationary points provably for all Lipschitz functions. Therefore, the current finite-time complexity bounds on finding $(\delta,\epsilon)$-stationary points cannot be directly transformed into a finite-time rate bound on $J(K^n)\rightarrow J^*$.

---

> > > ### Comment · Reviewer_fj7B · 2022-08-04
> > > **Updating score**
> > >
> > > Thanks to the authors for providing a very detailed response, and for their updates to the draft.
> > >
> > > I think the authors did a great job with the revision, and it has addressed my concerns regarding (mostly) end-to-end results for practical algorithms. I also appreciate the authors commenting on the subtleties of $(\delta, \epsilon)$-stationary points not implying close function values.
> > >
> > > In light of this, I am raising my score from 6 to 7. I think this is a nice paper that should be accepted.

---

### Official Review · Reviewer_Uf74 · 2022-07-09

**Rating:** 7
**Confidence:** 4
**Soundness:** 4 excellent
**Presentation:** 4 excellent
**Contribution:** 3 good

**Summary:**

This paper establishes asymptotic convergence for Goldstein's method for H-infinity policy search. On the plus side, it does so with elegant topological and optimization-theoretic arguments. A negative, however, is that the rates are non-quantitative, and it is not clear how the complexity of implementing the optimization would depend on problem parameters.

**Questions:**

1. Can the authors remark on the computational complexity of the sampling methods referred to in [49,13], and well as the zero-th order methods [9,10,31]? Do the authors conjecture that these methods could be implemented with samples scaling polynomially in relevant parameters?

2. Is there a way to bound the minimal separation between K_0 and K, Delta_0, in terms of a more natural problem quantity?

3. Can the argument of Theorem 2 be modified to given quantitative bounds for finite iterations?

**Limitations:**

As noted, the guarantees are non-quantitative and hence I am not sure if this is a good fit for NeurIPS.

However, I can be convinced! - if the authors can show precedent for similarly qualitative optimization papers (or if my fellow reviewers can), then I will be inclined to increase my score because I did enjoy this paper.

EDIT: After reviewing the author changes, most of my concerns regarding the computational infeasibility of finding approximate Goldstein differentials have been allayed. I am raising my review to a 7. I am tempted to give the paper an 8, but I am still concerned that the Delta_0 quantity is a little mysterious, and may hide nasty problem-dependent factors. Still the authors  should be proud of their excellent submission, I hope to see this in the program.

**Strengths And Weaknesses:**

I am torn about this paper; I thoroughly enjoyed reading it. I think it was excellently written, the proofs were clear and correct, and balanced concision with transparency perfectly. The fact that the results were purely qualitative made the paper easy and enjoyable to read. Were I asked to review this paper for an optimization or control journal, I would enthusiastically support acceptance.

I think the weakness of this paper is that, the NeurIPS theory community comes out of the learning theory tradition, where one cares about finite sample/iteration rates that reflect how problem parameters reflect the complexity of solving a problem. For example, many of the main papers on policy search for H2 control stress polynomial dependence in dimension, norms / bounds on various problem parameters, and on desired accuracy.

In this paper, the rates are purely asymptotic. Moreover, the algorithm requires the computation of a minimal norm element Goldstein differential, which may potentially be computationally prohibitive, especially if it requires sampling arbitrarily adversarial non-convex functions. Therefore, it is unclear to me whether even a single step of the proposed algorithm can be implemented efficiency (say, where the number of samples is poly(d,e,log(1/delta)). I did appreciate the authors discussion of the various implementation choices, and that level of specificity may be appropriate for other communities. But I think that for NeurIPS the author must convince the reader that the algorithm can be implemented efficiently in the sense that it can be performed in number of iterations / sampling calls which are polynomial in relevant problem parameters.

Similarly, the final guarantee in Theorem 2 is asymptotic, not giving a sense of rates.

EDIT: The authors have addressed the majority of my concerns, and I am increasing my score to a 7.

---

> ### Author Response · Authors · 2022-08-02
> **Response to Reviewer Uf74, Part I**
>
> Thanks for the constructive feedback. We really think that our paper fits NeurIPS well. As mentioned by Reviewer 5f2x, our work can be viewed as "a significant contribution to the interdisciplinary field of robust control and machine learning and a very meaningful initial step towards more general convergence results." The purpose of our paper is to bring more insights for understanding how direct policy search works on robust control problems. This topic is very relevant to the NeurIPS community. The reviewer's comment that it is preferred to include finite-time complexity bounds in NeurIPS papers makes sense to us. So we move the discussion on the finite-time complexity bound for finding $(\delta,\epsilon)$-stationary points from the supplementary material to the revised main paper (See Theorem 3 and 5 in the revised paper).  However, we do not think that every policy optimization theory paper in NeurIPS has to include an end-to-end sample complexity bound. For example, the following paper is published in NeurIPS 2020:
>
> [ZhangNeurIPS2020] K. Zhang, B. Hu, and T. Basar, On the Stability and Convergence of Robust Adversarial Reinforcement Learning: A Case Study on Linear Quadratic Systems, NeurIPS, 2020.
>
> The above paper does not include any sample complexity bounds. However, by studying the optimization landscape of the LQ adversarial RL setting, it brings important new insights for understanding the behaviors of policy-based adversarial RL methods. Our paper has a similar flavor. Our paper brings the following important insights:
>
> 1. One does not need to worry about local minimum when applying direct policy search to $\mathcal{H}_\infty$ state-feedback control problem since all the Clarke stationary points are global minimum.
>
> 2. One can address the nonconvex nonsmooth $\mathcal{H}_\infty$ robust control problem by combining the coercivity property and the descent property of the Goldstein subdifferential method.
>
> These are important insights for further studying the sample complexity of direct policy search on robust control problems, and hence we believe that our paper is a good fit for NeurIPS.  We also want to argue that it is quite common that optimization landscape, iteration complexity, and sample complexity are studied separately in different learning theory papers when it is difficult to study them together. This is very common for learning theory papers involving nonconvex optimization. In the nonsmooth setting, there is some fundamental difficulty preventing us to transfer the complexity bound for finding $(\delta,\epsilon)$-stationary points to a desired global convergence bound on $(J(K^n)-J^*)$. We will explain this more in our detailed response below.
>
> ***For example, many of the main papers on policy search for H2 control stress polynomial dependence in dimension, norms / bounds on various problem parameters, and on desired accuracy.***
>
> Such developments are reasonable for H2 control since the optimization problem involved in H2 control has nice structures. For more complicated robust control problems, the optimization is more difficult and hence one has to start from optimization landscape results which do not give sample complexity. This is exactly the case for [ZhangNeurIPS2020] which studies the LQ game subject to $\mathcal{H}_\infty$ constraints. For our current paper, the $\mathcal{H}_\infty$ control problem is more difficult than LQR due to the nonsmoothness. As mentioned before, the complexity for nonsmooth optimization of Lipschitz functions is quite subtle. The key subtlety is that  $(\delta,\epsilon)$-stationarity does not imply being $\delta$-close to an $\epsilon$-stationary point of $J$, and the finite-time complexity for finding $(\delta,\epsilon)$-stationary points does not imply too much about the convergence rate of the function values. Theorem 5 in Ref [58] of our revised paper even shows that there is no finite time algorithm that can find $\epsilon$-stationary points provably for all Lipschitz functions. Therefore, so far we can only show $J(K^n)$ will converge to $J^*$ without giving a rate. It is possible that one can develop finite-time bounds for $(J(K^n)-J^*)$ via exploiting other advanced properties of the $\mathcal{H}_\infty$ cost function. Then one can potentially build a complexity bound upon that result. However, this should be considered as a future task which is beyond the scope of our current paper.
>
> ***the rates are asymptotic.***
>
> Thanks for this comment. We move the discussions on the finite-time iteration complexity of finding $(\delta, \epsilon)$-stationary points to our revised main paper (see Theorem 3 and Theorem 5 in the revised paper). After Theorem 5, we also discuss how to interpret such results ($(\delta,\epsilon)$-stationarity does not imply being $\delta$-close to an $\epsilon$-stationary point of $J$ in the nonsmooth setting). As explained above, a finite-time bound on $(J(K^n)-J^*)$ is difficult in the nonconvex nonsmooth setting.

---

> > ### Author Response · Authors · 2022-08-02
> > **Response to Reviewer Uf74, Part II**
> >
> > ***it is unclear to me whether even a single step of the proposed algorithm can be implemented efficiently***
> >
> > In Section 4.2 of our revised main paper, we discuss four implementable algorithms: gradient sampling (GS), nonderivative sampling (NS), model-free NS, and Interpolated normalized gradient descent (INGD). All these methods can be efficiently implemented, and the required oracle types are discussed in Section 4.2 of the revised paper. Please see our revised paper. In addition, we have reported numerical results to show that these implementable variants work reasonably well in simulations.
> >
> > ***Can the authors remark on the computational complexity of the sampling methods referred to in [49,13], and well as the zero-th order methods [9,10,31]? Do the authors conjecture that these methods could be implemented with samples scaling polynomially in relevant parameters?***
> >
> > This is an open question. From simulations, these methods work reasonably well, and the behaviors seem to be consistent with the iteration complexity for finding $(\delta, \epsilon)$ stationary points.  However, more study will be needed to figure out the true sample complexity (the dependence on problem parameters).
> >
> > ***Is there a way to bound $\Delta_0$, in terms of a more natural problem quantity?***
> >
> > This is an interesting open question. Right now we are not aware of such a bound.
> >
> > ***Can the argument of Theorem 2 be modified to given quantitative bounds for finite iterations?***
> >
> > It depends on what type of finite-time iterations we are talking about. It can be modified to give a finite-time complexity bound on finding $(\delta,\epsilon)$-stationary points. However, such a bound cannot be directly transferred into a finite-time bound on $(J(K)-J^*)$. As mentioned before, there is some subtle difference between $(\delta,\epsilon)$-stationary points and $\epsilon$-stationary points for nonsmooth optimization of general Lipschitz functions. In our revised paper, we provide some explanations after we present Theorem 3 which gives the finite-time complexity of finding $(\delta,\epsilon)$-stationary points via the Goldstein subdifferential method.

---

> > > ### Comment · Reviewer_Uf74 · 2022-08-04
> > > **Thank you for adressing my concerns (some further suggestions)**
> > >
> > > Thank you for addressing my concerns. I appreciated the changes to the manuscript and have raised my score to a 7. I still would prefer some more insight on the magnitude of Delta_0 in typical situations, but I think the changes make the paper more than appropriate for the NeurIPS community.
> > >
> > > I did notice that the changes introduced a couple typos.
> > > 1. Lines 307-308 " for all n to ensure that the Goldstein subdifferential algorithm (13) to stay" - replace "to stay" with "stays"
> > > 2. Line 338: "Will be treated formally in the supplementary" - change supplementary to supplement.
> > >
> > > Last suggestion. It might be nice to reproduce [58, Theorem 8] as a formal statement in the appendix (just copy their statement and cite them). Then you can refer to that statement in the proof of Theorem 5.

---

### Official Review · Reviewer_5f2x · 2022-07-11

**Rating:** 7
**Confidence:** 3
**Soundness:** 3 good
**Presentation:** 3 good
**Contribution:** 3 good

**Summary:**

This paper considers the problem of direct policy search in the optimal $\mathcal{H}_\infty$ control framework. $\mathcal{H}_\infty$ control is an important framework in the robust control literature, but the problem of finding the optimal controller is challenging because it is a constrained nonconvex nonsmooth optimization problem. In this paper, the authors first study the landscape of the optimization problem to show the existence of a unique global minimizing set and any Clarke stationary points are global minimum. Then, they show a direct policy search method, the Goldstein subdifferential method, can stay in the feasible set and find a global optimal state-feedback controller.

**Questions:**

I encourage the authors to add a more detailed discussion about the future directions of this work. Despite a more general convergence bound, it will also be interesting to see if the linear time-invariant system and the time-invariant quadratic cost functions considered here can be generalized to more general dynamics/costs.

**Limitations:**

The authors discussed a limitation when practically implementing their algorithm in the last section, and proposed several potential solutions. I do not see any potential negative social impacts of this work.

**Strengths And Weaknesses:**

Strength: This paper makes a significant contribution to the interdisciplinary field of robust control and machine learning. Although similar results about the convergence of direct policy search exist in the literature of $\mathcal{H}_2$ control and mixed $\mathcal{H}_2/\mathcal{H}_\infty$ control, the authors discuss the difference between these settings clearly at the end of the Introduction. After reading the discussion, I believe the problem considered in this work is fundamental and challenging. This paper is also well-written and easy to follow in general.

Weakness: In Section 4.2, the authors mentioned that the minimum norm element of the Goldstein subdifferential $F^n$ can be difficult to evaluate in practice. However, the convergence result in Theorem 2 does not consider the potential error/noises when evaluating $F^n$ and perform the update. Further, Theorem 2 only provides an asymptotic convergence guarantee rather than a finite time error bound. Since the page limit is 9 pages, some contents in the appendix (e.g., the algorithm to evaluate $F^n$) can be moved into the main body.

In conclusion, I feel the strength of this work outweighs its weakness. This work is a very meaningful initial step towards more general convergence results.

---

> ### Author Response · Authors · 2022-08-02
> **Response to Reviewer 5f2x, Part II**
>
> ***I encourage the authors to add a more detailed discussion about the future directions of this work. Despite a more general convergence bound, it will also be interesting to see if the linear time-invariant system and the time-invariant quadratic cost functions considered here can be generalized to more general dynamics/costs.***
>
> We thank the reviewer for this useful suggestion. The complexity bound for finding $(\delta, \epsilon)$-stationary points can be directly extended to the nonlinear setting as long as the cost function is coercive. The proofs for Theorem 3 and Theorem 5 in the revised main paper only requires the cost to be coercive and locally Lipschitz. It is possible to prove this property or even enforce such a property via regularization for general nonlinear problems. Due to the space limit, we do not include this discussion for now. We will add it to the main paper later (if getting accepted) since one extra page is allowable for the final version.

---

> ### Author Response · Authors · 2022-08-02
> **Response to Reviewer 5f2x, Part I**
>
> We thank the reviewer for acknowledging our contributions. Detailed responses to your comments are provided below.
>
> ***In Section 4.2, the authors mentioned that the minimum norm element of the Goldstein subdifferential  can be difficult to evaluate in practice. However, the convergence result in Theorem 2 does not consider the potential error/noises when evaluating  and perform the update. Further, Theorem 2 only provides an asymptotic convergence guarantee rather than a finite time error bound. Since the page limit is 9 pages, some contents in the appendix (e.g., the algorithm to evaluate $F^n$) can be moved into the main body.***
>
> We thank the reviewer for this useful suggestion. We have moved some discussions on how to implement the Goldstein subdifferential method into the main body of our revised paper (see Section 4.2 of our revised paper). The discussions on the finite-time complexity of finding $(\delta,\epsilon)$-stationary points have also been moved back to the main body (see Theorem 3 and Theorem 5 in the revised main paper).
>
> On finite-time error bounds: There is some subtlety in understanding the complexity for nonsmooth optimization of Lipschitz functions. While the complexity result in Theorem 3 of our revised paper gives a reasonable characterization of the finite time performance of the Goldstein subdifferential method on the $\mathcal{H}_\infty$ state-feedback control problem, it does not quantify how fast $J(K^n)$ converges to $J^*$. Recall that $(\delta,\epsilon)$-stationarity means  $\delta$-Goldstein subdifferential includes an element with norm being smaller than or equal to $\epsilon$, while $\epsilon$-stationarity means the Clarke subdifferential includes an element with norm being smaller than or equal to $\epsilon$. It is well-known that  $(\delta,\epsilon)$-stationarity does not imply being $\delta$-close to an $\epsilon$-stationary point of $J$. Consequently, the function value of a $(\delta,\epsilon)$-stationary point can be very far from $J^*$ even for very small $\delta$ and $\epsilon$. Theorem 5 in [58] in shows that there is no finite time algorithm that can find $\epsilon$-stationary points provably for all Lipschitz functions. It is still possible that one can develop some finite time  bounds for $(J(K^n)-J^*)$ via exploiting other advanced properties of the $\mathcal{H}_\infty$ cost function. This is an important future task.
>
> On implementable variants of the Goldstein subdifferential method: In Section 4.2 of our revised main paper, we discuss four main algorithms: gradient sampling (GS), nonderivative sampling (NS), model-free NS, and Interpolated normalized gradient descent (INGD). Notice that GS, NS, and INGD are well-known variants of the Goldstein subdifferential method in the unconstrained setting. We mainly discuss how to adapt these algorithms for our constrained policy optimization setting, and some end-to-end theoretic results are added to strengthen our paper. Some key points from the added discussion are summarized as follows.
>
> 1. GS requires a first-order oracle to return $\nabla J(K)$ for any differentiable points in $\mathcal{K}$. From existing literature on GS, it is known that every cluster point of GS is Clarke stationary (with probability one). We can combine this result with our results (Lemma 4 and Theorem 1) to show that GS has an asymptotic global convergence guarantee. See Theorem 4 and a formal treatment in the supplementary material.
>
> 2. NS is the derivative-free version of GS, and only requires a zeroth-order oracle to return $J(K)$ for any points in $\mathcal{K}$. From existing literature on NS, it is known that every cluster point of NS is Clarke stationary (with probability 1). We can combine this result with our results (Lemma 4 and Theorem 1) to show that GS has an asymptotic global convergence guarantee. This is discussed in the revised version of our supplementary material.
>
> 3. Model-free NS: When the model is unknown, one has to estimate $J(K)$ using existing estimators for $\mathcal{H}_\infty$ norms. There exist many different options. Based on our experience, the MIMO power iteration method (from Ref[42] of our revised paper) works well for model-free implementation of NS. Currently, there is no sample complexity for model-free NS. Our numerical results in Figure 1 show that model-free NS closely tracks the convergence of its model-based counterpart given reasonable amount of samples.
>
> 4. INGD: This method also requires a first-order oracle to return $\nabla J(K)$ for any differentiable points in $\mathcal{K}$. The advantage of this method is that it yields a good finite-time complexity bound for finding $(\delta, \epsilon)$-stationary points. See Theorem 5 and a formal treatment in the supplementary material.
>
> Hopefully the above discussion convinces the reviewer that the Goldstein subdifferential method can be implemented for practical problems, and end-to-end theoretical results on these implementable variants can be derived.

---

### Official Review · Reviewer_HdMw · 2022-07-15

**Rating:** 7
**Confidence:** 3
**Soundness:** 3 good
**Presentation:** 3 good
**Contribution:** 3 good

**Summary:**

This paper studies the optimization landscape of the H_inf control problem. It is shown that all Clarke stationary points are global minimal even though H_inf is nonconvex. Further, a  Goldstein subdifferential method is proposed and shown to stay in the nonconvex feasible set and eventually reach a global optimal solution.

**Questions:**

Please add some numerical evaluation.

**Limitations:**

Yes

**Strengths And Weaknesses:**

H_inf control is an important robust control problem and is worth being studied, especially from the learning perspective. The optimization landscape results in this paper are significant. The Goldstein subdifferential method can guarantee the iterated control policies to stay in the nonconvex feasible set and eventually reach a global optimal solution. Further, the limitations of the Goldstein subdifferential method is discussed and implementable variants are proposed.

The major weakness of this paper is that no numerical evaluation is provided.

---

> ### Author Response · Authors · 2022-08-02
> **Response to Reviewer HdMw**
>
> ***The major weakness of this paper is that no numerical evaluation is provided.***
>
> We thank the reviewer for this useful suggestion. We have submitted a revised version of our paper and supplementary material which include numerical evaluations of various implementable forms of the Goldstein subdifferential method on several examples. Please see Section 5 in the main paper and Section D in the supplementary material. Our numerical results are consistent with our theory and show that direct policy search can find the global solutions of the $\mathcal{H}_\infty$ state-feedback synthesis problem.

---

### Public Comment · Authors · 2023-01-19
**Github link for Codes**

Repository: https://github.com/xi1ngang/Direct-Policy-Search-for-H-inf-State-Feedback-Problem

---

### Meta-Review · Area_Chair_iFxT · 2022-08-31

**Recommendation:** Accept
**Confidence:** Certain

**Metareview:**

The paper analyses direct policy search as an algorithm in H_inf control synthesis specifically for the problem of minimizing the H_inf closed-loop norm and show that it can achieve global optima. The authors observe that this is a constrained nonconvex nonsmooth optimization problem and by studying the landscape the landscape establish the existence of a unique global minimizing set and any Clarke stationary points are global minimum. They further propose using the Goldstein subdifferential method and show it can stay in the feasible set and find a global optimal state-feedback controller.

Overall the reviewers strongly appreciated the contribution and clarity of the paper. Some reviewers raised the concern of asymptotic vs non aymptotic results but were eventually satisfied by the author response, which highlighted bounds for delta, epsilon stationary points and the subtlety of delta, epsilon stationary points not being close to the global minima. There was unanimous agreement between reviewers that the paper would be a strong contribution to the conference and hence I recommend acceptance.

**Award:**

No

---

### Decision · Program_Chairs · 2022-09-14

Accept